# EgoThinker: Unveiling Egocentric Reasoning with Spatio-Temporal CoT

**Baoqi Pei**[1,2], **Yifei Huang**[1,3,*], **Jilan Xu**[1,4], **Yuping He**[5], **Guo Chen**[5],
**Fei Wu**[2], **Yu Qiao**[1], **Jiangmiao Pang**[1]
[1]Shanghai Artificial Intelligence Laboratory, [2]Zhejiang University,
[3]The University of Tokyo, [4]Fudan University, [5]Nanjing University
peibaoqi@gmail.com; hyf@iis.u-tokyo.ac.jp

## Abstract

Egocentric video reasoning centers on an unobservable agent behind the camera who dynamically shapes the environment, requiring inference of hidden intentions and recognition of fine-grained interactions. This core challenge limits current multimodal large language models (MLLMs), which excel at visible event reasoning but lack embodied, first-person understanding. To bridge this gap, we introduce **EgoThinker**, a novel framework that endows MLLMs with robust egocentric reasoning capabilities through spatio-temporal chain-of-thought supervision and a two-stage learning curriculum. First, we introduce **EgoRe-5M**, a large-scale egocentric QA dataset constructed from 13M diverse egocentric video clips. This dataset features multi-minute segments annotated with detailed CoT rationales and dense hand–object grounding. Second, we employ SFT on EgoRe-5M to instill reasoning skills, followed by reinforcement fine-tuning (RFT) to further enhance spatio-temporal localization. Experimental results show that EgoThinker outperforms existing methods across multiple egocentric benchmarks, while achieving substantial improvements in fine-grained spatio-temporal localization tasks. Full code and data are released at https://github.com/InternRobotics/EgoThinker.

## 1 Introduction

Humans possess a remarkable ability to reason, plan, and execute complex, goal-oriented behaviors within dynamic real-world environments. Recent works in Multimodal Large Language Models (MLLMs) have advanced the field of visual understanding [55, 1, 77, 54, 12]. Techniques such as chain-of-thought prompting [75, 89] and reinforcement fine-tuning [39, 27] further underscore the potential of MLLMs in high-level reasoning. However, existing approaches mainly address visual reasoning from an observer-centric, third-person viewpoint. This perspective fails to capture the embodied cognitive processes central to human reasoning, which naturally occur from the egocentric perspective.

Egocentric reasoning differs fundamentally from conventional visual reasoning due to the presence of the observer as an active participant in the scene. Thus, models must infer not only visible events but also the internal cognitive states, intentions, and future behaviors of the individual behind the camera. This reasoning process poses several unique challenges: (1) **Reasoning for complex tasks:** Understanding the rationale behind an action and predicting what comes next demands explicit cause-effect chains rather than isolated event recognition. (2) **Human–object interaction recognition:** Successful reasoning hinges on accurately localizing hands and manipulated objects.

---

*Corresponding author

39th Conference on Neural Information Processing Systems (NeurIPS 2025).

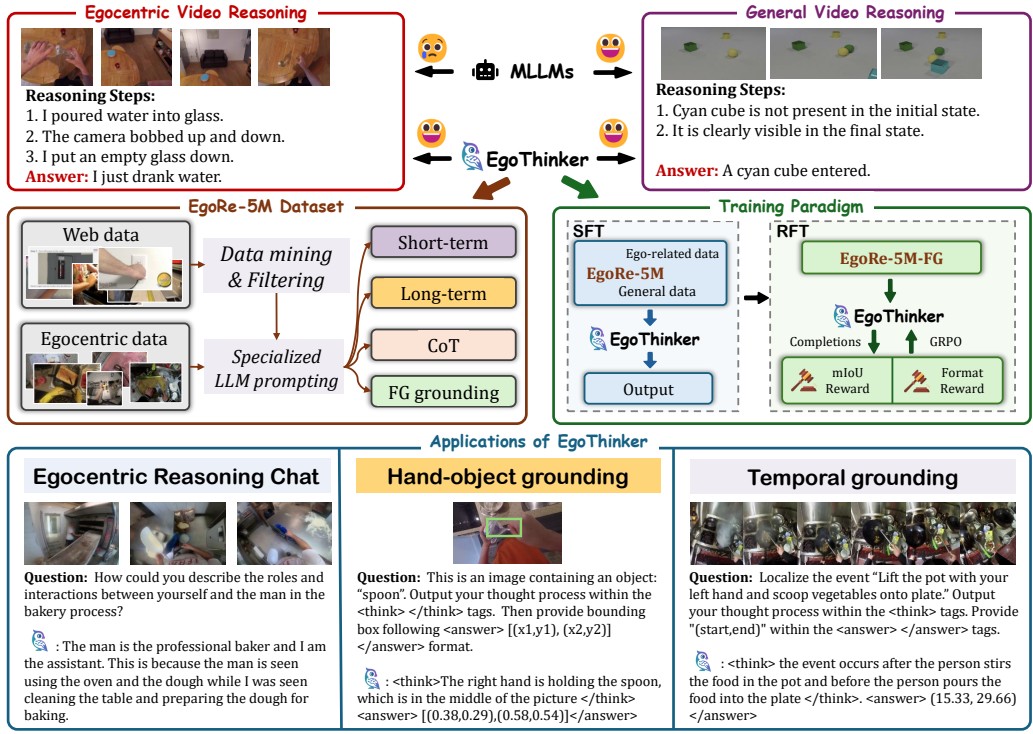

Figure 1: **Overview of our EgoThinker.** Unlike general video reasoning, egocentric video reasoning poses unique challenges because it must infer an unobservable camera wearer's interactions and intentions. EgoThinker addresses this by curating EgoRe-5M, a large-scale egocentric reasoning dataset, and applying a two-stage supervised and reinforcement fine-tuning paradigm. This design empowers robust egocentric reasoning chat, hand–object grounding, and temporal grounding, making EgoThinker a promising foundation for wearable assistants and embodied AI.

(3) **Multi-horizon temporal integration:** Egocentric video streams span minutes to hours, requiring models to track evolving context and retain fine-grained details across thousands of frames.

Existing egocentric datasets [97, 102] are mainly derived from Ego4D [25] and EgoExo4D [26]. They provide extensive collections of egocentric videos but lack explicit reasoning chains, temporally spanned annotations, and detailed fine-grained grounding data. Consequently, existing MLLMs [85, 101, 3], although successful in general visual understanding tasks, often struggle when reasoning about complex tasks in long-term egocentric videos.

In this work, we propose **EgoThinker**, a novel framework designed to enable robust egocentric reasoning in MLLMs. As illustrated in Figure 1, to overcome dataset limitations, we develop a pipeline to extract egocentric videos from large-scale web data to capture diverse real-world scenarios. Leveraging this, we construct EgoRe-5M, a large-scale egocentric QA dataset, featuring diverse questions spanning from seconds to several minutes. In the dataset, we incorporate detailed Chain-of-Thought (CoT) annotations to explicitly model the causal relationships underlying complex human activities, enabling models to emulate human-like causal inference and planning. Furthermore, recognizing the critical role of hand-object interaction in egocentric reasoning, we additionally introduce specialized data focusing on fine-grained spatio-temporal hand-object interactions.

Inspired by recent advances in reinforcement fine-tuning (RFT) [39, 27], EgoThinker employs a two-stage training strategy. Initially, the model undergoes supervised fine-tuning (SFT) [85, 101], utilizing EgoRe-5M to establish foundational understanding and reasoning. Subsequently, we employ RFT using spatio-temporal grounding data via GRPO [27] method. This paradigm significantly enhances the model's capability in fine-grained localization, temporal reasoning, and causal inference. Experiments demonstrate that EgoThinker outperforms existing state-of-the-art methods across multiple egocentric benchmarks, showcasing strong performance gains in egocentric QA [42, 21, 56], long-term video reasoning [15, 64], and spatio-temporal hand-object interaction localization [36, 19].

In summary, our contributions are: (1) EgoRe-5M, a large-scale egocentric QA dataset with chain-of-thought and hand-object annotations curated from diverse video sources. (2) A two-stage training regime combining supervised fine-tuning and reinforcement fine-tuning via GRPO to effectively couple high-level reasoning with low-level grounding. (3) EgoThinker, an MLLM setting new state-of-the-art on multiple egocentric video benchmarks, demonstrating coherent egocentric reasoning and precise spatial and temporal grounding.

## 2 Related work

**Egocentric Video Understanding.** Egocentric video understanding has recently garnered increasing research attention, with the introduction of large-scale egocentric datasets such as Ego4D. Previous works mainly focus on action understanding [72, 33, 24, 32, 34, 69, 29, 10], vision-language pre-training [56, 35, 108, 70] and hand-object interaction understanding [100, 9, 22, 93]. Recently, some methods [5, 97, 102, 30, 37] have attempted to use MLLMs for egocentric video understanding, and some targeting long-form egocentric videos [94, 87]. Different from these works, EgoThinker is the first model that enables reasoning and precise hand–object understanding in first-person videos.

**Multimodal Large Language Models.** Recent advancements in MLLMs [55, 1, 77, 31, 54, 12, 6] have shown robust comprehension and perception abilities. Video-LLaVA [55] and Videochat2 [51] enable MLLMs with general video understanding and temporal localization capability. Recent works extend to diverse domains including long-form video understanding [104, 11, 83, 44], robot learning [90, 45], and spatio-temporal perception [67, 62]. Chain-of-Thought (CoT) prompting [89] has proven effective in eliciting multi-step reasoning in LLMs. Recently, some works [13, 75, 14] use CoT techniques to enhance MLLM's visual reasoning capabilities. In our work, we construct large-scale QA samples with causal CoT captions to equip MLLMs with egocentric reasoning ability.

**Reinforcement Learning for MLLMs.** Recently, OpenAI-o1 [39] and DeepSeek-R1 [27] have shown that reward-driven fine-tuning can substantially enhance LLM reasoning. For MLLMs, many works [110, 59, 99, 20, 71, 60, 95, 103, 61] focused on leveraging reinforcement learning (RL) techniques with verifiable rewards to enhance visual reasoning capabilities with only a small amount of data. Videochat-R1 [52] enhances MLLM's temporal perception ability via RFT for general video understanding. Building on these advances, we construct fine-grained spatio-temporal grounding datasets and apply GRPO approach to endow MLLMs with precise hand–object localization and long-horizon causal inference in egocentric videos.

## 3 EgoThinker: Enhancing MLLMs Egocentric Reasoning Ability

In this section, we introduce our EgoThinker framework, designed to equip MLLMs with egocentric reasoning capabilities. We begin with the curation of EgoRe-5M, a large-scale egocentric instruction tuning dataset. We then describe how EgoRe-5M fuels our two-stage learning curriculum: initial supervised fine-tuning to establish foundational reasoning skills, followed by a reinforcement fine-tuning paradigm to sharpen hand-object grounding and temporal reasoning ability.

### 3.1 EgoRe-5M

Recent works [97, 102, 112] have constructed egocentric QA datasets but lack data for long-term causal reasoning and fine-grained spatio-temporal localization. This deficiency hinders the development of models capable of egocentric reasoning. To address this limitation, we propose EgoRe-5M, a large-scale egocentric QA dataset designed with rich causal reasoning and grounding data.

#### 3.1.1 Egocentric Video Collection

Accurate egocentric reasoning demands a vast and diverse collection of egocentric data. While existing datasets like Ego4D [25] and EgoExo4D [26] are valuable, their scale is notably smaller compared to web-sourced data. To overcome this scale bottleneck, we develop a multi-stage filtering pipeline to mine high-quality egocentric clips from web-sourced videos.

**(1) Web-scale Mining.** We choose the large instructional video dataset HowTo100M [65] as the data source in which numerous instructional video are recorded by a head-mounted or handheld

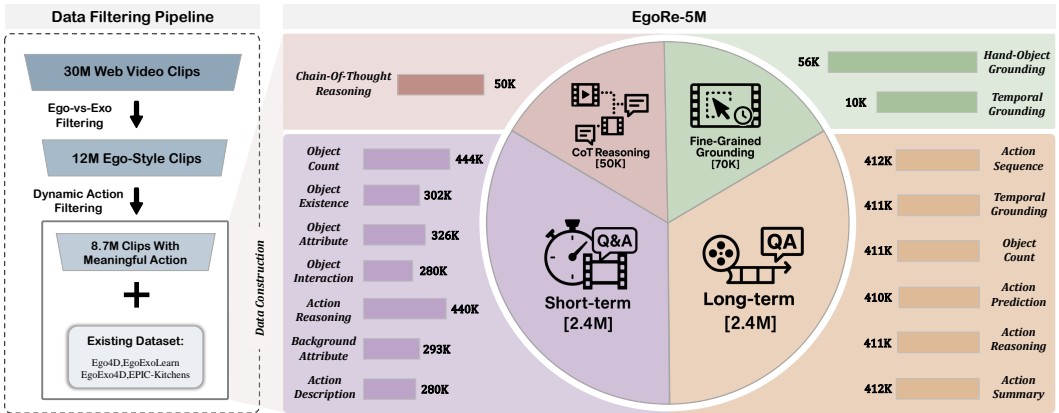

Figure 2: **Data Filtering Pipeline and EgoRe-5M Statistics.** With our multi-stage filtering pipeline, we construct EgoRe-5M, a large-scale QA dataset to facilitate egocentric reasoning in MLLMs.

camera featuring fine-grained hand-object interactions. Moreover, the task diversity and naturalistic narration within HowTo100M make it especially rich in egocentric perspectives. In particular, we select HTM-AA [28], a dataset containing a large number of temporally aligned video-narration pairs and Howto-Interlink7M [81], which additionally provides high-quality video caption annotations as our primary data sources, resulting in a total of 30M initial video clips. The duration of the video clips spans from a few seconds to several minutes.

**(2) Ego-vs-Exo Filtering.** To distinguish true egocentric footage from exocentric content, we train a classification model on balanced sets of manually labeled ego- and exo-centric clips. The classification model utilizes InternVideo backbone [85] followed by a two layer MLP. The model achieves 92% accuracy and 89% AUC on held-out validation data. Applying this model reduces our pool to 12M clips that exhibit clear first-person camera motion and range from seconds to minutes.

**(3) Dynamic Interaction Filtering.** Even after egocentric filtering, many clips remain static or depict group activities, offering limited value for reasoning about egocentric activities. Since most egocentric activities are centered on hand-object interactions, we run a pre-trained hand-object detector [74] to identify frames containing both a visible hand and an active object. We design several rules to filter the clips that exhibit hand-object interaction and dynamic changes, with each clip having a minimum duration of 2 seconds. This refinement yields 8.7M high-quality egocentric clips, each containing rich, dynamic interactions suitable for downstream QA annotation.

As a result, we combine our filtered 8.7M web data with existing egocentric datasets (Ego4D [25], EPIC-Kitchens [18], EgoExoLearn [36] and EgoExo4D [26]) to form a collection of 13M egocentric video clips in total. For details regarding the pipeline, please refer to the Supplementary A.

### 3.1.2 Egocentric Reasoning Data Construction

Prior egocentric QA datasets focus primarily on short video clips and raw narrations, falling short on long-term causal chains and spatial-temporal grounding, which are key components of egocentric reasoning. To address these gaps, we build EgoRe-5M, an automatically generated QA corpus containing four complementary task dimensions: short-term perception, long-term causal reasoning, chain-of-thought rationales, and fine-grained grounding. Figure 2 shows an overview of the data source of our EgoRe-5M. Since some video clips have no corresponding text annotations, or the text annotations come from low-quality automatic speech recognition, we employ Videochat2-HD [46], an efficient and robust video caption model, to annotate these videos with a sampling rate of 1 fps. Questions are formulated by specialized LLMs per split, detailed as follows.

**Short-term Data.**    To instill foundational egocentric perception skills, we generate a large-scale short-term QA split covering clips of 1–10 seconds. We design seven perceptual question types to capture immediate scene understanding: object existence, object attribute, object count, object interaction, action description, action reasoning and background attribute. For each clip, we combine the original text annotation with VideoChat2-HD captions and apply DeepSeek-V3 to instantiate and

answer these question templates. This process yields 2.4M QA pairs, ensuring broad coverage of objects, interactions, and immediate causal cues essential for downstream model pretraining.

**Long-term Data.** Human activities often progress through multiple steps, which makes long-form understanding essential for egocentric reasoning. To capture such extended causal chains, we aggregate consecutive clips into segments of 15–120 seconds and integrate their narrations into a single, coherent caption. We then design six question types to assess temporal and causal understanding within each segment: action sequence, temporal grounding, object count, action prediction, action summary and action reasoning. Utilizing DeepSeek-V3 on these concatenated captions, we automatically generate 2.5 million QA pairs. This aims to enhance models' abilities to connect events over long durations and deduce causal relationships.

**Chain-of-thought Data.** Recent advances in chain-of-thought (CoT) prompting [50, 107] have shown that explicitly conditioning LLMs on intermediate reasoning steps can significantly boost performance on complex inference tasks such as mathematics and coding. Visual CoT [75] extends this idea by supplying models with region-level hints to guide step-by-step reasoning. Thanks to the success of the open-source DeepSeek R1 [27] model, it is possible to utilize LLMs to generate reasoning processes. Similar to long-term data, we select video clips with dense captions and concatenate them to form new captions. Each description is then fed to DeepSeek R1, producing a question and an accompanying step-by-step rationale. We design a prompt to allow the model to decide whether a given segment warrants a CoT question, ensuring that only clips amenable to multi-step inference are annotated. The resulting split comprises 50K high-fidelity CoT QA pairs, each pairing a complex egocentric scenario with an explicit chain-of-thought process.

**Fine-Grained Grounding Data.** Hand–object interactions lie at the heart of egocentric reasoning. Existing specialized methods [100] perform well on these tasks, but existing MLLMs exhibit poor performance. To remedy this, we construct QA data for two complementary grounding tasks: *Hand-object Grounding* and *Temporal Grounding*.

First, leveraging EK-Visor's pixel-level masks for hands and active objects [19], we generate questions asking about the spatial positions of hands/objects. The input can take the form of an image. The model must first articulate its intermediate reasoning and then output a normalized bounding box. This split trains models to map hand and object visual cues to precise coordinates.

For temporal grounding, using EgoExoLearn's fine-grained temporal annotations [36], we pose questions that require selecting the exact time interval in a clip that contains the evidence needed to answer. Models need to provide step-by-step reasoning and output start–end times in seconds. This formulation emphasizes the ability to pinpoint moments of interest for downstream inference.

### 3.1.3 Data Statistics

As shown in Figure 2, our EgoRe-5M dataset comprises 5M question-answer annotations across four complementary splits. To verify the annotation quality, we randomly sample 500 QA pairs and check the correctness of answers and logical coherence of the intermediate rationales. Over 95% of reviewed samples meet our standards for accuracy and annotation quality. We provide qualitative examples in the supplementary to demonstrate the dataset's quality and range in object categories, action types, causal chains, and precise grounding scenarios.

### 3.2 Training EgoThinker

We employ a two-stage curriculum to imbue MLLMs with robust egocentric reasoning capabilities. First, we perform supervised fine-tuning on a carefully balanced mixture of general visual, egocentric, and QA datasets, including EgoRe-5M's short, long, and CoT splits, to establish core capabilities in object perception, causal inference, and multi-step planning (see details in the supplementary). Second, we refine spatio-temporal grounding via Reinforcement Fine-Tuning (RFT) using GRPO approach on EgoRe-5M's fine-grained grounding data. Together, these stages transform an off-the-shelf MLLM into our EgoThinker capable of egocentric reasoning.

Table 1: Overview of fine-tuning datasets used during training.

| Stage | Domain | Data Type | Amount | Training Dataset |
|-------|--------|-----------|--------|------------------|
| SFT | General | Caption | 100K | *Llavar[105], Sharegpt4o[17], Sharegpt4video[13], YC2[111], Webvid[84], Videochatgpt[63]* |
| | | VQA | 70K | *Next-QA [91], TVQA [48], Clevrer-QA [98], TGIF-QA [40], Star-QA [80]* |
| | Ego | Ego-Related | 390K | *SSV2 [109], Ego-QA [8], EgotimeQA [21]* |
| | | EgoRe-5M | 860K | *EgoRe-5M-Short, EgoRe-5M-Long, EgoRe-5M-CoT* |
| RFT | Ego | Grounding | 70K | *EgoRe-5M-FG* |

### 3.2.1 Supervised Fine-tuning

Supervised fine-tuning establishes the foundational reasoning, perception, and language skills of EgoThinker. To also preserve general visual understanding and conversational fluency, we assemble a 1.5M sample fine-tuning corpus drawn from a diverse mix of datasets of 4 categories as Table 1.

The details of the SFT training data are as follows: (1) **High quality caption datasets.** To maintain the general visual understanding and conversation ability, we select five high-quality datasets (llavar-gpt4-20k [105], sharegpt4o [17], sharegpt4video [13], webvid-2M [84], YouCook2 (YC2) [111] and videochatgpt [63]) containing image/video captions and conversational data. (2) **VQA datasets:** We use five commonly used QA datasets (Next-QA [91], TVQA [48], Clevrer-QA [98], TGIF-QA [40], and STAR-QA [80]) to sharpen QA skills. (3) **Ego-related datasets.** To emphasize the egocentric perspective understanding, we include existing ego-related datasets. Something-Something V2 (SSV2) [109] is relevant to action recognition and contain numerous ego-style clips. EgoTimeQA [21] and Ego-QA [51] are constructed from egocentric datasets, covering action recognition, and long-term video comprehension tasks. (4) **EgoRe-5M:** We sample from our EgoRe-5M short-term, long-term, and CoT splits to instill egocentric reasoning patterns. For details about SFT training, please refer to Supplementary B.

### 3.2.2 Reinforcement Fine-Tuning via GRPO

To enhance EgoThinker's spatio-temporal reasoning ability, we employ reinforcement fine-tuning (RFT) on the EgoRe-5M-FG split utilizing Group Relative Policy Optimization (GRPO) approach. Unlike traditional reward-model approaches [66], we adopt the rule-based scoring functions outlined in [47, 39, 76] to score the outputs of MLLMs directly. GRPO [27] then optimizes the policy by comparing groups of candidate responses without a separate critic network in.

**Preliminary.** Reinforcement Learning with Verifiable Reward (RLVR) replaces learned reward models with rule-based scorers that classify each output as correct or incorrect. DeepSeek R1-Zero's GRPO algorithm builds on RLVR by generating $N$ distinct candidate responses $o = \{o_1, \ldots, o_N\}$ for an input question $q$ through policy sampling. Each candidate is evaluated by a reward function to produce corresponding scores $\{r_1, \ldots, r_N\}$. To compare answers fairly, GRPO computes a normalized advantage for each response:

$$A_i = \frac{r_i - \text{mean}(\{r_i\}_{i=1}^N)}{\text{std}(\{r_i\}_{i=1}^N)}, \tag{1}$$

where $A_i$ represents the relative quality of the $i$-th answer. The policy update maximizes expected advantage-weighted likelihood, with a KL-divergence penalty to the reference model $\pi_{\text{ref}}$:

$$\max_{\pi_\theta} \mathbb{E}_{o \sim \pi_{\theta_{\text{old}}}(p)} \Big[ \sum_{i=1}^N \frac{\pi_\theta(o_i)}{\pi_{\theta_{\text{old}}}(o_i)} \cdot A_i - \beta \, \text{D}_{\text{KL}}\Big(\pi_\theta \, \| \, \pi_{\text{ref}}\Big) \Big], \tag{2}$$

where $\pi_\theta$ is the policy model, $\pi_{\text{ref}}$ is the reference model before optimization and $\beta$ is a regularization coefficient to control the KL-divergence. This objective encourages the model to allocate higher probability to top-ranked candidates while maintaining stability through KL regularization.

**Reward function design.** The design of the reward model is a critical step. We design two complementary rule-based rewards tailored to assist MLLM in egocentric reasoning.

*Format reward.* Following Deepseek-R1, we employ a format reward function to enforce the model to output the prediction value and thinking process in the specified format. Specifically, we expect the model to output its thinking process within `<think>...</think>` and the answer with

Table 2: **Results on egocentric video benchmarks.** For EgoTaskQA, we convert the dataset into a multiple-choice question format following [102].

| Method | EgoTaskQA | QAEgo4D | ERQA | EgoPlan-Val | EgoSchema | EgoMCQ | | VLN-QA | RES |
|---|---|---|---|---|---|---|---|---|---|
| | Acc. | Acc. | Acc. | Acc. | Acc. | Inter-Acc. | Intra-Acc. | Acc. | Acc. |
| LLaVA-Video-7B [106] | 55.0 | **68.6** | - | 39.8 | 57.3 | 86.7 | 37.3 | 32.0 | 31.1 |
| LLaVA-OneVision-7B [49] | 55.8 | 65.7 | - | 37.3 | 60.1 | 84.5 | 36.0 | 34.0 | 25.3 |
| VideoLLaMA3 [7] | 56.6 | 62.4 | - | 36.4 | 61.1 | 77.3 | 29.8 | 34.5 | 21.3 |
| VideoChat2-HD [51] | 45.5 | 52.0 | - | 35.7 | 55.8 | 87.1 | 36.4 | 43.0 | 24.5 |
| InternVL2-8B [16] | 61.0 | 66.4 | - | 34.2 | 64.2 | 79.0 | 31.0 | 46.0 | 30.0 |
| Exo2Ego-7B [102] | 48.1 | 62.1 | - | 42.7 | 61.3 | 88.4 | 41.2 | 44.5 | - |
| Qwen2-VL-7B [82] | 57.9 | 60.3 | 37.0 | 38.3 | 63.3 | 86.4 | 34.1 | 42.0 | 26.3 |
| **EgoThinker** | **64.4** | 66.2 | 41.8 | **47.1** | **67.6** | **89.3** | **41.4** | **54.0** | **39.5** |

`<answer>...</answer>`, and we design a format reward $R_{\text{format}}$. We utilize regular expression matching to determine whether the output of the MLLM matches our specified format. If matches, we assign $R_{\text{format}} = 1$; otherwise, we assign $R_{\text{format}} = 0$.

*IoU Reward in Spatio-Temporal Grounding.* The mIoU (mean intersection over union) metric offers clear guidance for the grounding task, making it an appropriate choice for the reward function. Therefore, using the fine-grained annotations from EgoRe-5M-FG, we calculate the spatial/temporal IoU between the predicted boxes/intervals and the ground-truth boxes/intervals as a reward function $R_{\text{o}_{\text{iou}}}/R_{\text{t}_{\text{iou}}}$. Thus, for hand-object grounding task, we get $R_{\text{og}} = R_{\text{format}} + R_{\text{o}_{\text{iou}}}$, and for temporal grounding task, we get $R_{\text{tg}} = R_{\text{format}} + R_{\text{t}_{\text{iou}}}$.

During RFT, we use the combined rewards to strengthen EgoThinker's ability to generate well-structured reasoning traces and accurate spatial and temporal groundings.

## 4 Experiments

The main experiments are all conducted based on Qwen2-VL-7B [82]. To train EgoThinker, we first perform SFT on our curated datasets, then apply RFT via GRPO.

**Benchmarks.** To thoroughly assess performance, we organize our evaluation benchmarks as follows: *(1) Egocentric Benchmarks.* We evaluate core first-person reasoning capabilities on six established datasets: EgotaskQA [42], QAEgo4D [21], EgoPlan [15], EgoSchema [64], EgoMCQ [56], ERQA [78] and VLN-QA [51]. *(2) Cross-View Skill Transfer (RES).* To gauge the model's ability to generalize learned skills across perspectives, we introduce the Referenced Egocentric Skill (RES) benchmark of 4-way MCQs using paired clips from EgoExoLearn [36] and EgoExo4D [26]. *(3) Fine-grained Spatio-Temporal Grounding.* We assess spatial and temporal grounding on two newly constructed specialized HOI benchmarks derived from EK-Visor [19] and EgoExoLearn [36]. *(4) General Video Understanding.* Finally, to confirm that EgoThinker maintains broad applicability, we report results on three general video benchmarks including MVBench [51], Perception Test [2], and VideoMME [23]. Please refer to the Supplementary for details about these benchmarks.

### 4.1 Quantitative Evaluation of Egothinker

**Egocentric Benchmarks.** Table 2 shows a comprehensive comparison of EgoThinker against our baseline Qwen2-VL-7B and other leading MLLMs on the egocentric benchmarks. EgoThinker establishes new state-of-the-art performance across all tasks, achieving a 4.4% absolute gain on EgoPlan, 3.4% on EgoSchema, and 8.0% on VLN-QA. This clearly demonstrates EgoThinker's strong capacity for long-horizon planning, semantic inference, and goal-oriented question answering in egocentric video. In contrast, baseline models exhibit inconsistent strengths: LLaVA-Video leads on QAEgo4D while InternVL2 excels on EgoSchema and EgotaskQA, revealing an apparent lack of generalization in existing MLLMs to egocentric perception and reasoning.

On the Referenced Egocentric Skill (RES) cross-view benchmark, where prior MLLMs perform at near-random levels, EgoThinker outperforms the second best model by 8.4%, underscoring its unique ability to transfer learned skills between perspectives. These results confirm that our EgoThinker effectively unleashes the egocentric reasoning capabilities of MLLMs.

Table 3: EgoThinker results on hand-object and temporal grounding tasks.

| Method | EK-Visor | | EgoExoLearn | |
| --- | --- | --- | --- | --- |
| | mIoU | Loc-Acc. | mIoU | R1@0.05 |
| LLaVA-Video [106] | 46.7 | 67.7 | 1.30 | 7.8 |
| Qwen2VL-7B [82] | 56.7 | 64.5 | 1.53 | 5.4 |
| Qwen2.5VL-72B [79] | 64.1 | 71.7 | 21.1 | 49.9 |
| EgoThinker | **53.7** | **80.3** | **25.2** | **63.9** |

Table 4: EgoThinker results on general video benchmarks.

| Method | MVBench | Perception Test | VideoMME |
| --- | --- | --- | --- |
| MiniCPM-V2.6 [96] | 67.1 | 58.1 | 60.9 |
| InternVL2 [16] | 66.4 | 60.1 | 54.0 |
| LLaVA-Video [106] | 58.6 | 67.9 | 63.3 |
| InternVideo2.5 [86] | **74.0** | **76.2** | **65.1** |
| Qwen2VL [82] | 68.2 | 70.3 | 62.9 |
| EgoThinker | 70.0 (+1.8) | 72.7 (+2.4) | 62.9 (+0.0) |

**Spatio-Temporal Grounding Benchmarks.** For spatial grounding of hand–object, we measure mean Intersection over Union (mIoU) and Localization Accuracy (Loc-Acc). For Loc-Acc, we determine the correctness by checking whether the predicted box's center is within the ground truth box. As shown in Table 3, off-the-shelf 7B-parameter MLLMs achieve only modest mIoU and Loc-Acc on EK-Visor. Scaling up to Qwen2.5-VL-72B improves these scores; however, EgoThinker, after our two-stage training paradigm, surpasses this baseline by a wide margin.

For temporal grounding, we employ the temporal window's mIoU and R1@0.05 as metrics. In Table 3, results in EgoExoLearn show that 7B-parameter MLLMs perform at near-zero levels, indicating almost no temporal localization capability. Qwen2.5-VL-72B improves the performance, but still underperforms our EgoThinker. These improvements confirm that our training paradigm effectively equips MLLMs to reason about "when" and "where" in dynamic egocentric video.

**General Benchmarks.** We further evaluated EgoThinker on three standard video understanding benchmarks. As shown in Table 4, EgoThinker exhibits no degradation across any of these tasks, matching or exceeding the Qwen2-VL-7B baseline. Notably, it achieves clear gains on MVBench and Perception Test, underscoring that our two-stage paradigm not only unlocks robust egocentric reasoning but also enhances broad video understanding capabilities.

## 4.2 Ablation Studies and Discussions

Table 5: **Ablations on EgoRe-5M.** We use different splits of the data for training to validate the effectiveness of our dataset.

| Method | Training Data | | | | EgoTaskQA | QAEgo4D | Egoschema(val) | EK-Visor | |
| --- | --- | --- | --- | --- | --- | --- | --- | --- | --- |
| | Short | Long | CoT | FG | Acc. | Acc. | Acc. | mIoU | Loc-Acc. |
| Baseline | - | - | - | - | 57.7 | 60.3 | 68.2 | 28.6 | 64.5 |
| +SFT | ✓ | - | - | - | 61.6 | 63.1 | 69.1 | 29.1 | 64.9 |
| | ✓ | ✓ | - | - | 64.2 | 63.7 | 71.1 | 28.9 | 64.5 |
| | ✓ | ✓ | ✓ | - | 64.3 | **67.2** | **71.9** | 28.5 | 64.4 |
| +RFT | ✓ | ✓ | ✓ | ✓ | **64.4** | 66.1 | 71.8 | **53.7** | **80.3** |

**Training data.** To systematically evaluate the contribution of each component in EgoRe-5M dataset, we perform ablations by applying SFT on the short-term, long-term, and CoT splits, while employing RFT on the fine-grained split. As shown in Table 5, the short-term split boosts performance on three QA benchmarks especially EgotaskQA, highlighting the value of dense, scenario-diverse clips. The long-term split improves EgoSchema as expected, while incorporating the CoT split leads to marked enhancements in QAEgo4D, a dataset focusing on episodic-memory-based question answering. This shows that explicit causal reasoning traces can benefit memory-driven tasks. Finally, the FG split significantly enhances spatial and temporal grounding. Overall, these results verify that each split in EgoRe-5M provides essential, complementary information for different aspects of egocentric reasoning, and that jointly leveraging them enables more holistic understanding of first-person activities.

**Comparison of SFT and RFT.** Table 6 presents a direct comparison of SFT and RFT on the EgoRe-5M-FG split. We can observe that compared to SFT, utilizing RFT significantly enhances the model's

performance across all tasks. Additionally, utilizing SFT affects performance on Egoschema and QAEgo4D, whereas RFT maintains robust performance across these benchmarks. This demonstrates that our method not only enhances spatio-temporal perception but also preserves the model's ability to generalize across diverse egocentric tasks.

Table 6: Ablations on different training paradigms.

| Stage | Method | Data | EK-Visor | | Egoexolearn | | EgoSchema(val) | QAEgo4D |
|---|---|---|---|---|---|---|---|---|
| | | | mIoU | Loc-Acc. | mIoU | R1@0.05 | Acc. | Acc. |
| **Stage1** | Baseline | - | 28.6 | 64.5 | 1.53 | 5.4 | 68.2 | 60.3 |
| | + SFT | SFT Data | 28.5 | 64.4 | 1.81 | 7.0 | **71.9** | **67.2** |
| **Stage2** | Stage1 + SFT | EgoRe-5M-FG | 38.9 | 74.1 | 9.84 | 24.9 | 71.4 | 62.1 |
| | Stage1 + RFT | EgoRe-5M-FG | **53.7** | **80.3** | **25.2** | **63.9** | 71.8 | 66.1 |

**Input Frame Number.** We investigate the impact of input sequence length by adjusting the number of sampled frames from 1 to 64 during inference, with the results presented in Figure 3. Across all benchmarks, it is evident that as the number of frames increases, the performance improves progressively. Notably, a marked decline in performance is observed when only a single frame is used, especially for EgoSchema. This indicates that long-term reasoning is an intrinsic need in egocentric reasoning.

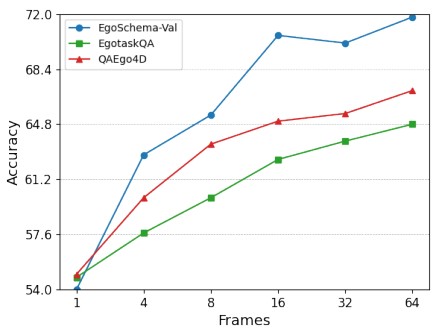

Figure 3: Ablations on number of frames.

We investigate the impact of input sequence length by adjusting the number of sampled frames from 1 to 64 during inference, with the results presented in Figure 3. Across all benchmarks, it is evident that as the number of frames increases, the performance improves progressively. Notably, a marked decline in performance is observed when only a single frame is used, especially for EgoSchema. This indicates that long-term reasoning is an intrinsic need in egocentric reasoning.

**Ablations on hallucination detection.** To validate whether our model exhibits hallucinations, we conduct evaluations on two widely-adopted benchmarks: (1) VideoHallucer [88] for assessing MLLMs, which contains object-relation, temporal, semantic detail and extrinsic hallucination detection. (2) POPE (Polling-based Object Probing) [53] for object hallucination detection.

The experimental results in Table 7 demonstrate a marginal performance decrease (0.6%) on VideoHallucer, while achieving a significant improvement of 3.2% on the POPE benchmark. We attribute this differential performance to our model's enhanced hand-object grounding capability, which particularly enhances its robustness against object hallucination.

Table 7: Results on hallucination detection benchmarks.

| Method | VideoHallucer | POPE |
|---|---|---|
| Qwen2VL [82] | 47.6 | 83.6 |
| EgoThinker | 47.0 | 86.8 |
| GPT4o | 53.3 | - |

**Grounding Data and Egocentric Video Understanding.** To find the relationship between the spatio-temporal grounding data and egocentric video understanding, we conduct further analysis on these benchmarks. We found that benchmarks such as EgoTaskQA, QAEgo4D, EgoSchema, and EgoPlan focus on high-level understanding, planning, and reasoning. They only require the identification of simple objects and do not involve fine-grained interaction details. Our baseline model, Qwen2-VL, already possesses a certain level of capability, which explains why our grounding data did not yield significant improvements.

Previous works [68, 43] demonstrated that using hand and object as supervisory signals enhances egocentric video understanding abilities. Therefore, we select three more fine-grained benchmarks: ERQA, EgoMCQ and EGTEA for evaluation.

As shown in Table 8, the inclusion of grounding data significantly improved model performance across all benchmarks. These datasets involve tasks directly related to our hand-object grounding data, such as hand/robotic arm motion, object identification, and object classification.

Table 8: Ablations on the grounding data.

| Method | ERQA | EgoMCQ | EGTEA |
|---|---|---|---|
| Qwen2-VL | 37.0 | 86.4/34.1 | 32.4 |
| EgoThinker(without grounding data) | 40.1 | 87.6/38.3 | 33.1 |
| EgoThinker | 41.8 | 89.3/41.4 | 35.4 |

## 4.3 Qualitative Results

In Figure 4, we show the hand–object grounding and reasoning traces for 4 methods. Our baseline Qwen2-VL performs poorly on both hand and object grounding. GPT-4o [38] generates a coherent chain-of-thought but misidentifies the target object (mislabeling the knife) and inaccurately localizes the hand. Grounding-DINO [58] is an expert model specialized in object grounding, however, it cannot distinguish left from right hand. In contrast, EgoThinker first approximates the object location and then outputs an accurate bounding box after thinking, demonstrating its superior fine-grained spatial reasoning in egocentric contexts.

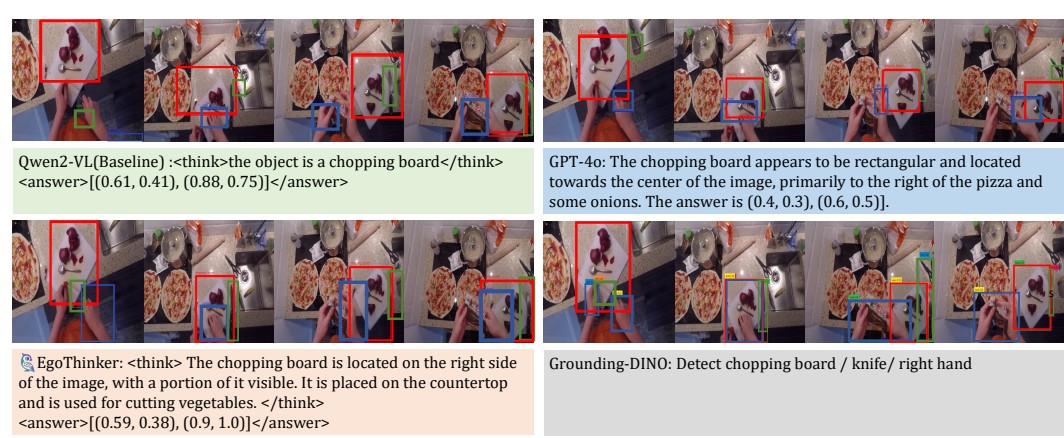

Figure 4: Hand-object grounding visualization on EK-Visor dataset. We compare our method to baseline Qwen2-VL, GPT-4o and expert model Grounding-DINO. We utilize different prompts tailored to each model and for each image, we use "chopping board", "knife", "right hand" as query for grounding.

## 5 Conclusion

We introduce EgoThinker, a framework toenhance egocentric reasoning in MLLMs. By constructing EgoRe-5M, a large-scale egocentric instruction-tuning dataset, we provide the rich, structured data required for human-like spatio-temporal understanding and reasoning. To efficiently leverage our data, we employ the SFT-RFT combined paradigm to further equip EgoThinker with robust causal planning, long-horizon context integration, and precise localization capabilities. Experimental results demonstrate that EgoThinker achieves state-of-the-art performance across multiple egocentric benchmarks and significantly improves performance on spatio-temporal perception tasks, while maintaining general video understanding abilities. Despite these advantages, EgoThinker possesses certain limitations, such as reliance on extensive annotations and offline fine-tuning, and cannot perform real-time inference in resource-constrained environments. Future work will focus on real-time adaptation, richer multimodal integration, and self-supervised learning to further enhance its robustness and efficiency and we hope to extend EgoThinker into the field of Embodied AI, enabling more interactive agent behaviors in real-world environments.

**Acknowledgement** This work is funded in part by the National Key R&D Program of China (2022ZD0160201), JSPS KAKENHI JP25K24384, and Shanghai Artificial Intelligence Laboratory.

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

# A  Details About EgoRe-5M

**Video Source.**    After obtaining our filtered 8.7M video clips from web data, we combine them with existing egocentric datasets including Ego4D [25], EPIC-Kitchens [18], EgoExoLearn [36], and EgoExo4D [26] to form a comprehensive dataset totaling 13 million clips. Specifically, we utilize 4M clips from  [73] within Ego4D. For EPIC-Kitchens, we select 56,000 frames from the EK-Visor dataset. From EgoExoLearn, we carefully curated 10,000 L1-level video clips for temporal grounding training and 400 samples for the RES benchmark. Regarding EgoExo4D, we incorporated 500 samples for the RES benchmark evaluation.

**Dynamic Interaction Filtering.**    Since many video clips remain static or depict group activities, we need to filter out irrelevant clips for egocentric reasoning. After obtaining object bounding boxes for each frame using a hand-object detector, we design the following filtering rules to select video clips with dynamic interaction:

*Step 1:* For all video clips, we first examine the hand bounding boxes. If the number of hands detected exceeds two ($N_{hands} > 2$) indicating multi-person activities, we discard such clips:

$$\text{Filter Clip} \Longleftrightarrow N_{hands} > 2 \tag{3}$$

*Step 2:* For the remaining clips, given a clip $C$ with $N$ frames, we set a threshold $\alpha = 0.7$. The clip is discarded if the total number of object bounding boxes is less than $\alpha \times N$:

$$\text{Filter Clip} \Longleftrightarrow \sum_{i=1}^{N} N_{objects}^{(i)} < \alpha \times N \tag{4}$$

*Step 3:* After the first two steps, we obtain clips containing single-person hand-object interactions. To further select dynamic clips, we calculate the maximum inter-frame hand displacement. For clip $C$, we compute the hand center $(x_t, y_t)$ in each frame $t$. The clip is kept if the maximum center displacement exceeds 10% of the image size ($min(H, W)$):

$$\text{Filter Clip} \Longleftrightarrow \sum_{t1=1,t2=1}^{H,W} \sqrt{(x_{t_1} - x_{t_2})^2 + (y_{t_1} - y_{t_2})^2} > 0.1 \times min(H, W) \tag{5}$$

Through these three steps, we filter out high-quality dynamic video clips, which can be used to train our egocentric reasoning MLLM.

**Data Statistics.**    Table 9 presents the details of EgoRe-5M, including 16 question types and corresponding QA examples for each type. Notably, in the Chain-of-Thought (CoT) split, we provide detailed and meaningful reasoning processes to the question. For the Fine-Grained split, we introduce two special tokens, `<think>` and `<answer>`, to structurally format the reasoning process and final answers, which facilitates the execution of our reward function during model training.

# B  Training

**SFT Data.**    To balance training efficiency and model performance, we carefully curated our training dataset as shown in Table 1. While using the complete dataset would lead to prohibitive computational costs and performance degradation due to data imbalance, We filter each dataset : for video caption dataset, we select 170K samples on total; for ego-related dataset, we select 390k QA samples in total; for our EgoRe-5M, we select 810K samples, including 410K from short-term splits, 400K from long-term split and 50K from CoT split.

**Training Details.**    We use QwenVL2-7B as our baseline for training. For SFT, we adopt $max\_pixels = 200704$, $min\_pixels = 3136$, $lr = 1e - 6$, $epoch = 1$ for training. We utilize 32 A100 GPUs and train for 30 hours. For RFT, we adopt $lr = 1e - 5$, $epoch = 1$ for training. We utilize 8 A100 GPUs and train for 12 hours. Notably, during RFT training phase, we first train hand-object grounding task and then train temporal grounding task.

| Data Split | Question Type | Number | Example |
|---|---|---|---|
| **Short-term (0-10s)** | Object Existence | 302K | *What object is the person interacting with in the video?* 
 The person is interacting with a grass-trimming tool. |
| | Object Attribute | 326K | *What is the state of the garlic during the slicing process?* 
 The garlic is being sliced and held in place by the left hand. |
| | Object Count | 444K | *How many people are in the video?* 
 There are three people in the video. |
| | Object Interaction | 280K | *What is the chef doing with the white soup bowl?* 
 The chef is rinsing the white soup bowl under running water to ensure it is thoroughly clean. |
| | Action Description | 280K | *What actions are being performed by the hands in the video?* 
 The left hand moves downward to grasp an object, while the right hand pulls a handle to the side. |
| | Action Reasoning | 440K | *Why does the left hand move to the center of the frame?* 
 The left hand moves to the center of the frame to pick up a book. |
| | Background Attribute | 293K | *What is the setting or background of the video?* 
 The background appears to be an outdoor area with grass, likely a lawn or garden. |
| **Long-term (15-120s)** | Action Sequence | 412K | *What is the sequence of actions performed by the left and right hands when cutting and placing mango pieces?* 
 The left hand moves downward to place the mango on the fruit cutter while the right hand holds the cutter steady. Then, the left hand stabilizes the mango while the right hand cuts it. after cutting, the left hand moves downward to place the mango piece while the right hand remains still. |
| | Temporal Grounding | 411K | *When does person z demonstrate a clear preference for using their left hand?* 
 Person z demonstrates a clear preference for using their left hand when they remove the cap and place it on the table, which occurs between 7.33s to 10.25s. |
| | Object Count | 411K | *How many objects are being interacted with across the video clips?* 
 The objects being interacted with include an art brush, paint, an art board, a piece of paper, tissue, and a paint palette, totaling six objects. |
| | Action Prediction | 410K | *After person c places a piece of cloth into the bag, what is the likely next action?* 
 The likely next action is person c sorting the clothes in the bag. |
| | Action Summary | 411K | *What are the key actions involving the left hand across the video clips?* 
 The key actions involving the left hand include person y picking up the phone, person z removing and placing the cap on the table, person z picking up a camera from person k, and the left hand moving in various directions while interacting with objects. |
| | Action Reasoning | 412K | *Why does the person use both hands to apply plaster to the ceiling?* 
 The person uses both hands to ensure precision and control while applying plaster. One hand scoops and guides the plaster, while the other maneuvers the scraper, allowing for an even and smooth application on the ceiling. |
| **CoT (15-200s)** | Chain-of-thought Reasoning | 50K | *How did the person adjust their hand positioning while operating the lawnmower?* 
 As described in 'Person uses both hands to lift a shovel'), then dug into the soil with it ('Person digs into the soil with a shovel, using both hands'). After digging, he spread the soil using his left hand while steadying a container with his right ('Man A spreads soil with his left hand, while his right hand steadies the container'). Finally, after completing the task, he carefully placed the shovel on the grass by guiding it down with his left hand and steadying the handle with his right ('Man A carefully places the shovel on the grass, his left hand guiding it down while his right hand steadies the handle'). This sequence shows a logical workflow: lifting the tool, executing the primary action (digging) distributing the material, and safely storing the tool afterward. |
| **Fine-grained Grounding** | Temporal Grounding | 10K | *To accurately pinpoint the event "[QUESTION]" in the video, you need to identify a time interval from which the answer to the question can be deduced. Output your thought process within the <think></think>tags. Then, provide the start and end times (in seconds, precise to two decimal places) in the format "(start,end)" within the <answer></answer>tags.* |
| | Hand-Object Grounding | 56K | *This is an image containing an object: "[OBJECT]" ,and output the bounding box of this object in the image. Output your thought process within the <think></think>tags. Then provide your bounding box within the <answer></answer>tags,following <answer>(x_min,y_min),(x_max,y_max) </answer>format. The bounding box coordinates are normalized to the range [0, 1], relative to the width and height of the image.* |

Table 9: **Statistics of the proposed EgoRe-5M.** The table shows 4 data splits and 16 question types, along with the corresponding example question-answer pairs. More details can be found in Section 3.1.2.

## C Benchmark Details

**EgoTaskQA.** EgoTaskQA [42] is a large-scale egocentric video questionanswering dataset designed to evaluate models' understanding of goal-oriented human tasks. It is derived from LEMMA dataset [41], focusing on aspects such as action effects, intent, multi-agent collaboration, and object interactions. The dataset emphasizes reasoning types including spatio-temporal understanding, causal dependencies, and task planning, supported by 30K annotated state transitions. It includes a variety of question formats, such as binary and open-ended queries, to ensure a balanced and unbiased evaluation. To evaluate this dataset, we reformulate the original open-ended QA samples into a multiple-choice question through a systematic conversion process. Specifically, we first aggregate all potential answers into a list. For each question, BERT [4] is used to compute semantic similarity scores between the ground-truth answer and all candidate answers in the pool. The four most semantically similar answers were then selected to construct the new multiple-choice question, with the ground-truth answer serving as the correct option.

**QAEgo4D.** QAEgo4D represents a specialized benchmark for assessing episodic memory through video-based question answering. This dataset, derived from the Ego4D, measures the ability of vision-language models to comprehend and reason about dynamic visual sequences. Each entry consists of four key components: (1) an egocentric video clip, (2) a manually crafted question, (3) its corresponding answer, and (4) precise temporal localization of the relevant visual evidence. To ensure annotation quality, the dataset employs redundant textual descriptions that undergo cross-verification. QAEgo4D provides researchers with a robust framework for investigating memory-related video understanding tasks. To evaluate the dataset, we select the closed-set QA split parsed by [92].

**EgoPlan.** EgoPlan serves as a multimodal benchmark for evaluating human-like planning abilities in AI systems through egocentric video understanding. Derived from large-scale egocentric datasets including Epic-Kitchens and Ego4D, the benchmark comprises 4,939 rigorously validated multiple-choice questions, spanning 3,296 distinct task objectives and 3,185 executable action sequences across 419 diverse real-world environments. By simulating real-world decision-making scenarios, the benchmark facilitates progress in multimodal reasoning for practical planning applications. In our experiments, we adopt the dataset's predefined validation split for evaluating planning performance, as ground-truth annotations for the test set remain undisclosed.

**EgoSchema.** EgoSchema represents a novel benchmark framework for assessing long-form video comprehension in multimodal AI systems. Derived from the Ego4D video corpus, this evaluation suite comprises 5,000+ meticulously annotated multiple-choice question-answer pairs, sourced from 250+ hours of unscripted daily human activities captured in real-world settings. The benchmark presents a unique challenge where AI models must analyze three-minute video clips and select the most accurate response from five plausible alternatives, testing their capacity for sustained visual-temporal reasoning and contextual understanding.

**EgoMCQ.** EgoMCQ is a multiple-choice question-answering dataset designed to assess video-text alignment in egocentric vision systems. Derived from Ego4D, it includes 39,000 questions based on 468 hours of egocentric video covering a wide range of human activities. Each question involves selecting the correct video clip from five options based on a narration, with two settings: "inter-video", for distinguishing between different videos, and "intra-video", for fine-grained context within the same video.

**VLN-QA.** VLN-QA represents a specialized evaluation benchmark for assessing multimodal navigation understanding in indoor environments through question-answering tasks. Derived from the VLN-CE framework, this dataset comprises thousands of carefully annotated multiple-choice items paired with egocentric video sequences that replicate authentic navigation scenarios. The benchmark specifically examines a system's ability to interpret visual-spatial information and correlate it with textual queries. For our implementation, we utilize the preprocessed dataset version established in VideoChat2's experimental setup.

**RES.** To validate our model's cross-view reasoning capability, we developed a Cross-View Skill Transfer Benchmark named RES (Referenced Egocentric Skill). RES leverages paired exocentric–egocentric clips from the EgoExoLearn and EgoExo4D datasets. Each example presents one exocentric video as a reference and four candidate egocentric clips, and the model must identify which egocentric view corresponds to the reference. This multi-choice protocol rigorously tests the ability to transfer observed skills across perspectives. The final benchmark comprises 936 curated samples. Although RES is crafted to validate EgoThinker's cross-view reasoning, we anticipate it will become a valuable resource for the broader embodied AI community.

**Grounding Benchmark.** For the grounding benchmark construction, we select existing annotations to derive our evaluation dataset. Specifically, for the hand-object grounding benchmark, we curated our dataset from EK-Visor, which provides bounding box annotations. Our methodology involved extracting bounding boxes from segmentation masks in the validation set, serving as ground-truth references. This process yielded a comprehensive collection of 13,000 object queries for evaluation purposes. For the temporal grounding task, we strategically selected the EgoExoLearn dataset due to its unique dual-level annotation structure, which makes it suitable for temporal localization tasks. We select L1-level (coarse-grained) video clips as our primary video sources and L2-level (fine-grained)

temporal windows as precise ground truth annotations. To this end, we curate an evaluation set of 3,000 test instances.

## D    Additional Experiments

**Effects Of Extra Video Sources.**    Table 10 provides a comparative analysis of model performance with and without the inclusion of QA samples sourced from the HowTo100M dataset. The results indicate consistent performance improvements across all evaluated benchmarks when leveraging the HowTo100M-derived data, with particularly notable gains on long-term understanding tasks such as QAEgo4D and EgoPlan. We attribute these improvements to the long-term split in EgoRe-5M, which is primarily derived from HowTo100M, significantly enhancing the model's capacity for extended temporal reasoning. These findings underscore the effectiveness of our data curation strategy in enabling robust egocentric reasoning ability.

Table 10: Ablations on our EgoRe-5M. We evaluate the impact of incorporating data filtered from the HowTo100M dataset on performance.

| Data | EgoTaskQA | QAEgo4D | EgoPlan-Val | VLN-QA |
| --- | --- | --- | --- | --- |
| | Acc. | Acc. | Acc. | Acc. |
| Baseline | 57.9 | 60.3 | 38.3 | 42.0 |
| w/o Howto100M | 62.2 | 61.6 | 41.3 | 50.0 |
| w Howto100M | **64.4** | **66.2** | **47.1** | **54.0** |

**Results On General Grounding Task.**    To further validate EgoThink's grounding capabilities, we conduct additional experiments on the COCO dataset [57]. As evidenced by Table 11, our model demonstrates significant performance improvements despite never being trained on COCO data, which substantiates its strong generalization ability for object grounding tasks.

Table 11: Results on COCO detection dataset.

| Method | testA | testB |
| --- | --- | --- |
| | mIoU | mIoU |
| Qwen2VL-7B | 34.1 | 33.6 |
| EgoThinker | 55.2(+21.1) | 57.8(+24.2) |

**Additional Visualization Results.**    Figure 5 compares temporal grounding outputs for the baseline, Gemini 2.5-Pro [77], and EgoThinker on a sample query. We observe that Qwen2-VL fails to identify the query and lacks a reasoning process in both examples. Gemini2.5-Pro provides dense video captions and can accurately align the captions with the corresponding events. However, the experimental results reveal that the model exhibits inferior localization performance when processing scenes with hand-object interactions. EgoThinker combines an explicit reasoning trace with precise temporal window prediction, accurately identifying the start and end times of the queried event. These visualizations underscore EgoThinker's unique ability to deliver both interpretable reasoning and robust spatio-temporal grounding in first-person video.

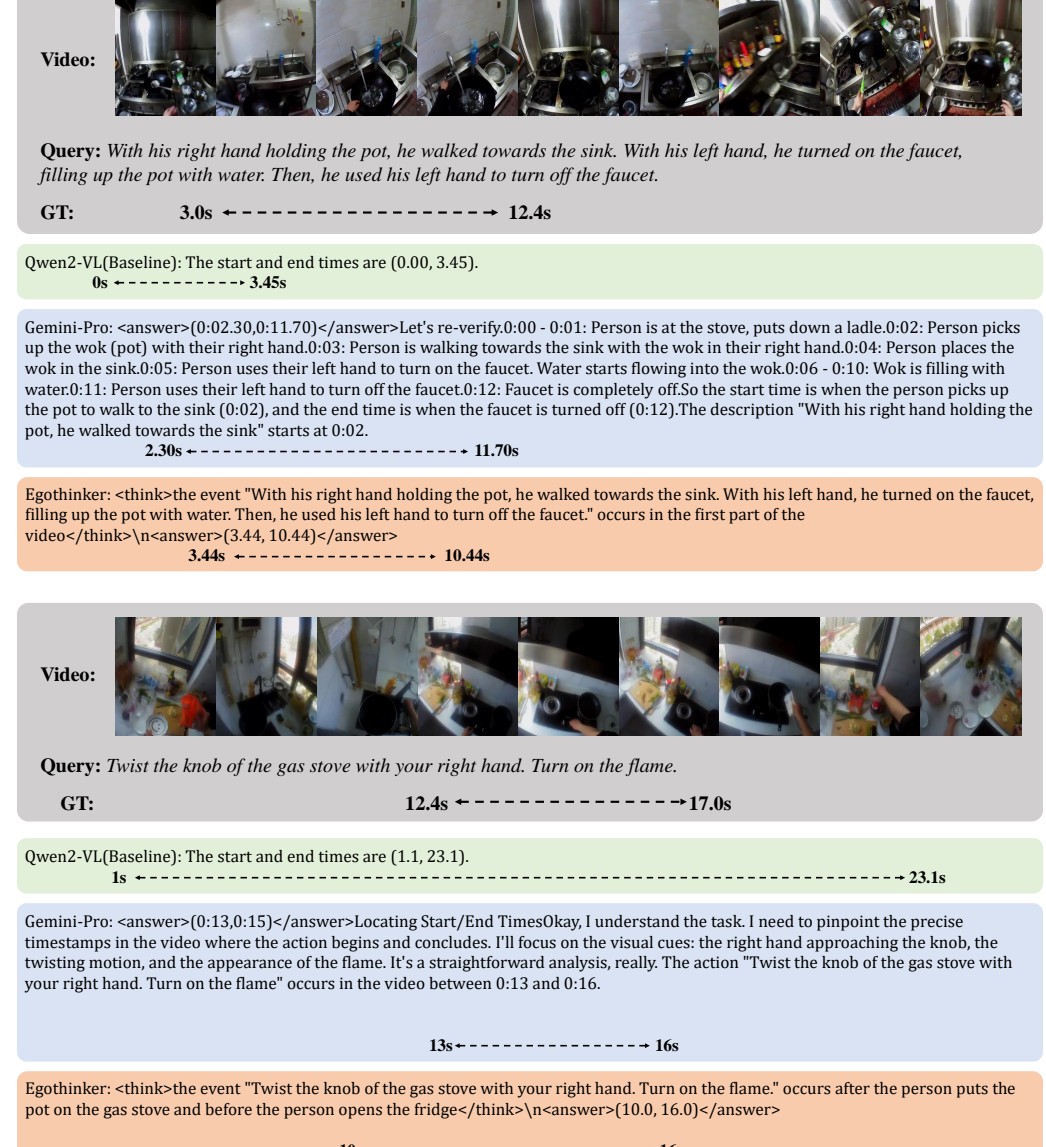

Figure 5: Temporal grounding visualization on the EgoExoLearn dataset. We compare our method to baseline Qwen2-VL and one of the strongest MLLM Gemini2.5-Pro.

