# OpenReview forum: "EgoThinker: Unveiling Egocentric Reasoning with Spatio-Temporal CoT"
_NeurIPS.cc/2025/Conference — NeurIPS 2025 poster_

### Official Review · Reviewer_k3DC · 2025-06-21

**Clarity:** 3
**Significance:** 2
**Originality:** 2
**Rating:** 4
**Confidence:** 4

**Summary:**

Current multimodal large language models (MLLMs) excel at reasoning about observable events but struggle with embodied egocentric understanding. To bridge this gap, this paper introduces EgoRe-5M, a dataset encompassing four complementary task dimensions. Leveraging this dataset, they first equip the model with fundamental egocentric video understanding and reasoning abilities via supervised fine-tuning (SFT) on short-term, long-term, and chain-of-thought (CoT) data. Subsequently, they apply reinforcement learning on fine-grained grounding tasks to further enhance the model’s spatiotemporal localization capabilities. Experimental results across multiple egocentric benchmarks demonstrate that the proposed approach significantly improves both basic egocentric video understanding and fine-grained spatiotemporal localization.

**Questions:**

1. The process by which 13 million egocentric video clips were converted into 5 million QA pairs is insufficiently detailed. A more transparent and systematic description is needed to clarify this transformation pipeline.
2. The introduction of the proposed benchmarks, such as EK-Visor and EgoExoLearn, lacks sufficient detail. The paper should provide a clearer rationale for their necessity and include evidence that these benchmarks are unbiased, representative, and appropriate for evaluating spatiotemporal grounding.
3. Table 5 does not adequately demonstrate the contribution of fine-grained grounding tasks to egocentric video understanding. This weakens the central claim made early in the paper, that successful reasoning depends on accurately localizing hands and manipulated objects. A more thorough analysis is needed to substantiate this connection.

**Ethical Concerns:**

["NO or VERY MINOR ethics concerns only"]

**Final Justification:**

The rebuttal has satisfactorily addressed most of my concerns. I would like to revise my score to BA.

**Limitations:**

For certain datasets, such as QaEgo4D and EgoExoLearn, the use of EgoTimeQA and the inclusion of fine-grained grounding data effectively place the evaluation in a fine-tuned setting. However, zero-shot performance would be more convincing for demonstrating egocentric video understanding capabilities. Therefore, I recommend conducting additional zero-shot experiments and providing a clearer explanation of the current evaluation protocol.

**Quality:**

4

**Strengths And Weaknesses:**

1. The paper includes comprehensive ablation studies that effectively demonstrate the contributions of different data components and the effectiveness of the two-stage training strategy.
2. This work involves extensive data collection and processing, which requires substantial effort and resources.
3. The first claimed contribution of this paper is the construction of the EgoRe-5M dataset. Its distinction from traditional egocentric video datasets such as MM-Ego [1] primarily lies in the inclusion of CoT data and grounding data. However, the CoT annotations are mainly generated by DeepSeek R1 without a specific design tailored to the characteristics of egocentric videos. On the other hand, Table 5 shows that the introduction of grounding data has no significant impact on egocentric video understanding, yet the paper does not clearly explain how these results relate to the connection between grounding data and egocentric video understanding. These suggest a lack of deep understanding of the fundamental differences between egocentric and exocentric perspectives.
4. The second contribution is the application of a two-stage training regime combining SFT and reinforcement learning(RL) for model training. Similarly, this approach does not incorporate considerations specific to egocentric videos. Compared to UniVG-R1 [2], the main difference lies only in the task setting. Although the work is built upon egocentric video datasets and tasks, the methods are equally applicable to exocentric videos and thus do not make a distinctive contribution to advancing the field of egocentric video understanding.
5. The two main contributions of this paper can essentially be summarized as the application of the CoT-SFT+RL paradigm to egocentric video tasks. Recent studies have introduced task-specific enhancements to RL methods, such as the Difficulty-Aware Weight Adjustment Strategy proposed in UniVG-R1 [2] and the AVA-GRPO approach presented in Vad-R1 [3]. In contrast, this paper only modifies the data and task settings, without providing deeper insights into how the RL paradigm could be adapted or improved to better address the unique challenges of egocentric video understanding.

[1] Ye H, Zhang H, Daxberger E, et al. MMEgo: Towards building egocentric multimodal LLMs for video QA.
[2] Bai S, Li M, Liu Y, et al. Univg-r1: Reasoning-guided universal visual grounding with reinforcement learning.
[3] Huang C, Wang B, Wen J, et al. Vad-R1: Towards Video Anomaly Reasoning via Perception-to-Cognition Chain-of-Thought.

---

> ### Author Rebuttal · Authors · 2025-07-31
>
> We thank the reviewer for insightful questions and feedback. Due to the word limit, we will simplify the description in the rebuttal.
>
> ### **Details of the QA Generation Process**
>
> 1.We first use a Dynamic Interaction Filtering Process. Due to the word limit, please refer to Supplementary A for details.
>
> 2.After filtering, we construct the QA data.
> In Section 3.1.2, we provide a detailed description of the formatting and procedures for different data splits. We believe your primary concern is that the selection of clips is not clear, particularly for long-term questions. We add additional details below.
>
> Our source long videos contain hundreds of clips. For generating long-term and CoT QA pairs, we employ the following protocol:
>
> (1) We sequentially scan each video's clips, randomly selecting 30-120s segments (30% for 30-60s, 70% for 60-120s).
>
> (2) We concatenate all clips within each selected window, discarding segments with <5 clips (due to insufficient captions).
>
> After compiling long-term QA clips, we remove duplicates from the short-term subset to create our final clip collection. We will add the complete detail to the revision.
>
> ### **Details of Our Benchmarks**
>
> 1. EK-Visor:
> This dataset provides hand-object segmentation annotations, ensuring the quality of bounding boxes. It covers over 200 representative scenes with diverse hand-object interactions, and it is the most widely used dataset for egocentric spatial grounding. It addresses a critical gap in standard benchmarks (e.g., RefCOCO for images, NExT-GQA[1] for videos), which lack egocentric interaction data.
>
> 2. EgoExoLearn: There are very few benchmarks for egocentric temporal grounding. Current benchmarks, such as Ego4D-NLQ[2], require locating a temporal window from a natural language query. However, these queries often suffer from poor quality and inconsistent lengths. EgoExoLearn contains the largest number of high-quality temporal boundary annotations. Its annotations are generated by trained human annotators, ensuring the data is high-quality, unbiased, and representative. Furthermore, the dataset provides timestamped annotations at coarse and fine-grained levels, making it ideally suited for building a temporal grounding benchmark to evaluate a model's ability to localize egocentric events.
>
> ### **CoT annotations has no specific design tailored to the characteristics of egocentric videos.**
> For the selection of clips for our CoTs, we utilized captions from the Ego-HOD[3] dataset. These captions augment high-level semantic information with the hand and object interactions and movements. This ensures that our source information is inherently rich with unique egocentric cues.
>
> Building on this, our CoT generation process is context-aware. It is guided by temporally localized questions grounded in the egocentric clips, ensuring that the reasoning traces are linked to egocentric cues such as hand-object interaction, temporal progression, and egocentric intent. As a result, our CoT annotations contain information for both fine-grained and multi-horizon reasoning, which is crucial for egocentric videos.
>
> ### **Relationship Between Grounding Data and Egocentric Video Understanding**
> We thank the reviewer for the suggestion and conducted further analysis on these benchmarks. We found that benchmarks such as EgoTaskQA, QAEgo4D, EgoSchema, and EgoPlan focus on high-level understanding, planning, and reasoning. They only require the identification of simple objects and do not involve fine-grained interaction details. Our baseline model, Qwen2-VL, already possesses a certain level of capability, which explains why our grounding data did not yield significant improvements.
>
> Previous works[3,4,5] demonstrated that using hand and object as supervisory signals enhances egocentric video understanding abilities. Therefore, we selected three more fine-grained benchmarks, ERQA, EgoMCQ and EGTEA for evaluation.
>
> | Model    | ERQA | EgoMCQ | EGTEA |
> |---------|---------|---------|---------|
> | Qwen2-VL | 37.0 | 86.4/34.1 | 32.4 |
> | EgoThinker(without grounding data) | 40.1 | 87.6/38.3 | 33.1 |
> | EgoThinker | 41.8 | 89.3/41.4 | 35.4 |
> | Gemini1.5 | 41.8 | - | - |
>
> As shown in the table, the inclusion of grounding data significantly improved model performance across all three benchmarks. These datasets involve tasks directly related to our hand-object grounding data, such as understanding hand/robotic arm motion, object identification, and object classification.
>
> To provide a more intuitive demonstration of the grounding data's effectiveness, we present inference examples below:
>
> (1) Q:How do you need to rotate the dumbbell for it to fit back in the weight holder?
>
> A:'To determine how to rotate the dumbbell for it to fit back in the weight holder, we first need to understand the orientation of the dumbbell and the weight holder. **Grounding of the Key Object**: The dumbbell is the key object here. It is shown lying on its side with the handle pointing upwards. The weight holder is positioned above the dumbbell, with its base facing downwards.. **Current Position**: The dumbbell is currently lying horizontally on the floor, with its handle pointing upwards.
> Thus, you need to rotate dumbbell counter-clockwise 90 degrees and move it above to fit back in the weight holder.
>
> (2) Q:The right hand holding a blue object is in motion. What is it nearest to?
>
> A:The gripper is nearest to the stove top burner. The key objects: left hand, right hand,the stove top burner, and the blue object. The right hand is described as holding the blue object, which is located near the stove top burner. Since **the right hand is moving towards the stove top burner**, it must be closest to that area. During the process, the left hand does not move. Therefore, the right hand is nearest to the stove top burner.
>
> These examples demonstrate that accurate grounding of objects and hands and reasoning about their relationships is essential for deriving correct answers. As discussed in our paper, current MLLMs exhibit limited capability in distinguishing hands, underscoring the critical role of precise visual grounding in such tasks.
>
> In summary, grounding data provides a more pronounced benefit in egocentric scenarios that require fine-grained understanding, such as object classification, action recognition, and embodied reasoning in robot manipulation.
>
> ### **The Contribution of Our Training Paradigm**
> We thank the reviewer for highlighting these recent works in RFT and prompting us to clarify our contributions to the training paradigm.
>
> Our primary goal with RFT was to enhance two crucial skills for egocentric reasoning without degrading the general reasoning abilities.
> Unlike previous works which mainly optimize for a single task, our framework simultaneously optimizes for spatial and temporal grounding. Our investigation into this multi-task challenge revealed a critical challenge: mixed training caused unstable updates and slow convergence, requiring over 3k steps.
>
> We address this through proposing a simple yet effective **single-task batching** strategy, where each training batch contains data from only one grounding task, alternated throughout training. This approach stabilized the learning process, accelerating convergence to 1k steps and leading to improved final performance on both grounding benchmarks.
>
> This insight offers practical value for multi-task RFT applications, and we will ensure it is included in the final version to provide a more complete description of our methodology. We agree that further exploration into multi-task RL optimization is a promising direction for future work.
>
> We appreciate the reviewer pointing us to UniVG-R1 and Vad-R1. We would also like to respectfully note that these papers are concurrent work with our submission, as both were released publicly as pre-prints very recently. Therefore, while they represent exciting and parallel advancements in the field, our work was developed independently and addresses the distinct challenge of multi-task egocentric grounding.
>
> ### [Limitations] Zero-shot evaluation
> We thank the reviewer for this important feedback.
> For EgoTimeQA, we identify that it may share identical information with QAEgo4D, indicating potential data contamination issues. To ensure evaluation fairness, we excluded EgoTimeQA and conducted retraining.
>
> | Model    | QAEgo4D | EgoTaskQA | EgoSchema | EgoMCQ |
> |---------|---------|---------|---------|---------|
> | Qwen2-VL | 60.3 | 57.9 | 63.3 | 86.4/34.1 |
> | EgoThinker | 66.2| 64.4 | 67.6 | 89.3/41.4 |
> | EgoThinker w/o EgoTimeQA | 64.5 | 64.6 | 67.4 | 89.0/41.5
>
> As the table shows, removing EgoTimeQA resulted in a 1.7% drop on QAEgo4D, confirming shared QA patterns. In zero-shot setting, our model still outperforms the baseline by a significant margin of 4.2%, demonstrating its robust reasoning capabilities.
>
> For the evaluation protocol, we conduct zero-shot experiments in MCQ format for QA benchmarks. Regarding spatial-temporal grounding tasks, evaluation is performed after RFT on a small training set.
> Due to the lack of suitable grounding benchmarks in the egocentric domain, we validate our approach through grounding-relevant benchmarks including ERQA, QAEgo4D, and EgoPlan(Table 2 in the paper). Additionally, we provide results between SFT and RFT in temporal grounding to demonstrate the effectiveness of our RFT paradigm.
>
> | Model    | mIoU | R1@0.05 |
> |---------|---------|---------|
> | SFT | 14.3 | 42.8 |
> | RFT | 25.2 | 63.9 |
>
> [1] Xiao et al, Can i trust your answer? visually grounded video question answering.
>
> [2] Ramakrishnan et al, Naq: Leveraging narrations as queries to supervise episodic memory.
>
> [3] Pei et al, Modeling fine-grained hand-object dynamics for egocentric video representation learning.
>
> [4] Zhang et al, Helping hands: An object-aware ego-centric video recognition model.
>
> [5] Xu et al, Do Egocentric Video-Language Models Truly Understand Hand-Object Interactions.

---

> > ### Author Response · Authors · 2025-08-04
> >
> > Dear reviewer,
> >
> > We appreciate your time and effort in evaluating our work. As the discussion stage is ending soon, we wonder if our response answers your questions and addresses your concerns? Thanks again for your very constructive and insightful feedback!

---

> > ### Comment · Reviewer_k3DC · 2025-08-04
> >
> > Thank you for the responses! It has addressed most of my concerns.

---

### Official Review · Reviewer_Fem7 · 2025-06-30

**Clarity:** 3
**Significance:** 3
**Originality:** 2
**Rating:** 4
**Confidence:** 3

**Summary:**

This paper introduces EgoThinker, a framework designed to enhance egocentric reasoning in MLLMs. Its core contributions include:
- EgoRe-5M：A large-scale egocentric QA dataset;
- Two-stage training：SFT on EgoRe-5M for foundational understanding and reasoning, followed by RFT using GRPO to improve spatio-temporal grounding.

The empirical results demonstrate that EgoThinker achieves state-of-the-art performance across multiple egocentric video benchmarks

**Questions:**

Is it possible to leverage RFT to improve performance on tasks other than Spatio-Temporal Grounding? Specifically, could this method be adapted or extended to enhance long-horizon planning, semantic inference, or goal-oriented question answering?

**Ethical Concerns:**

["NO or VERY MINOR ethics concerns only"]

**Final Justification:**

After considering the authors’ rebuttal and clarifications, I find that several concerns, especially around data quality assurance, were satisfactorily addressed. The responses on dataset novelty provided a reasonable distinction from prior work, though the incremental nature of the contributions remains. The discussion on adapting RFT to other tasks was informative but remains speculative without empirical evidence. Overall, the paper is technically solid, presents a substantial dataset and training framework, and demonstrates strong empirical results, but the originality is moderate. Given the balance of strengths over remaining limitations, I maintain the borderline accept rating of 4.

**Limitations:**

Despite mentioning manual screening of the data, there remains a lack of rigorous quality assurance measures during the dataset construction phase. This could potentially impact the reliability and generalizability of the results obtained from EgoThinker. Further details on quality control mechanisms would strengthen the robustness of the study.

**Quality:**

3

**Strengths And Weaknesses:**

### Strengths:
The authors built a large dataset (EgoRe-5M) for egocentric reasoning in MLLMs and validated a two-stage training method (SFT+RFT), showing significant improvement in Hand-object Grounding. Extensive comparative experiments against leading models demonstrate EgoThinker's effectiveness.

### Weakness
While EgoRe-5M and the two-stage training approach are notable, they suffer from a lack of novelty, as they represent incremental rather than groundbreaking innovations in dataset creation and fine-tuning strategies. Similar efforts in data creation:

Ji, Yuheng, et al. "Robobrain: A unified brain model for robotic manipulation from abstract to concrete." Proceedings of the Computer Vision and Pattern Recognition Conference. 2025.

Cheng, Sijie, et al. "Videgothink: Assessing egocentric video understanding capabilities for embodied ai." arXiv preprint arXiv:2410.11623 (2024).

---

> ### Author Rebuttal · Authors · 2025-07-31
>
> We thank the reviewer for insightful questions and feedback. We believe that we can address your concerns in the rebuttal.
>
> ### **[Weaknesses] Contributions of Our Dataset.**
>
> While our dataset, EgoRe-5M, shares some similarities with existing datasets, our core contributions are distinct and address key limitations in the field.
>
> First, we address the challenge of limited video sources in the egocentric domain. For instance, RoboBrain's data is exclusively sourced from the Open-X-Embodiment[1] dataset, while VidEgoThink's video content is derived entirely from the GoalStep[2] subset of Ego4D. This limited video sources limit the model's generalization capabilities. In contrast, our work introduces a pipeline to curate data from large-scale web data, significantly broadening the scope of training data.
>
> Second, our dataset encompasses a broader spectrum of question types and achieves data scaling. While RoboBrain focuses on action planning and VidEgoThink includes four dimensions of questions, EgoRe-5M further incorporates complex reasoning and temporal grounding tasks. Furthermore, we have achieved a significant scale-up in data volume (5k in VidEgoThink vs 5M in our case), which is crucial for training more powerful egocentric understanding models.
>
> ### **[Questions] Adapting RFT for Other Tasks.**
> We agree that adapting RFT to other tasks is a feasible direction. The critical element lies in the design of the reward function, which must be tailored to the specific task objectives. For example, for tasks such as long-horizon planning or semantic inference, we can employ a large language model (e.g., GPT-4o) to score the generated outputs with the ground truth, using these scores as the reward signal. For goal-oriented QA, the task could be framed as a classification problem to supervise the model's selection of the correct target. These are relatively intuitive approaches. We believe that the design of more sophisticated reward functions, as well as the exploration of joint multi-task RFT, are valuable and promising avenues for future research.
>
> ### **[Limitations] Data Quality Assurance.**
> To ensure the quality and reliability of our dataset, we conducted several steps to quantify:
>
> 1.**CLIP-based Alignment Quality Analysis.** We analyze the alignment quality between the caption and the video via the CLIP score. Due to the large scale of the dataset, we sampled 100K instances to compute the score. Specifically, we chose ViCLIP[3] pre-trained on a large amount of video data, and compared it against the original concatenated captions as a baseline. We found that the score of our CoTs is 71.4, while the original caption score is 74.7.
>
> While the score of our data is lower than the descriptive caption baseline, this is an expected outcome. The source captions describe concrete, visible objects and actions, which aligns perfectly with CLIP's training objective. In contrast, our data contains reasoning steps, causal links, and inferred goals (e.g., “to align the dumbbell, a rotation is needed”), which do not always have a direct visual correlate.  A score of 71.4 still indicates strong semantic alignment, confirming that our annotations are highly relevant to the video's content.
>
> 2.**Hallucination detection.** To validate whether EgoThinker exhibits hallucinations, we conducted evaluations on two widely-adopted benchmarks: (1) VideoHallucer[4] for assessing large video-language models, which contains object-relation, temporal, semantic detail and extrinsic hallucination detection. (2) POPE (Polling-based Object Probing)[5] for object hallucination detection.
>
> | Model    | VideoHallucer | POPE |
> |---------|---------|---------|
> | Qwen2-VL | 47.6 | 83.6 |
> | EgoThinker | 47.0 | 86.8 |
> | GPT4o | 53.3| - |
>
> The experimental results demonstrate a marginal performance decrease (0.6%) on VideoHallucer, while achieving a significant improvement of 3.2% on the POPE benchmark. We attribute this differential performance to our model's enhanced object grounding capability, which particularly enhances its robustness against object hallucination.
>
> 3.**Manual Review.** We conducted extensive manual verification on the most complex Chain-of-Thought (CoT) data split to validate our data quality. Based on the length of the generated CoT, we selected 5k samples from our CoT data split for manual review. The goal of this scoring was to assess whether the generated caption contained any non-existent objects or action descriptions when compared to the video. We set up a 4-point scale to categorize hallucination severity: Severe Hallucination, Moderate Hallucination, Partial Hallucination and No Hallucination. We get 4068, 577, 279, 76 for score = 3,2,1,0. We observe that hallucinations primarily manifest as the model outputting non-existent goals or actions based on the provided captions, with hardly any instances of non-existent objects. Furthermore, the majority of captions exhibit no hallucinations or only minor ones, indicating good alignment quality in our CoT generation.
>
> If you have any additional questions or suggestions, please feel free to tell us.
>
> [1] Open x-embodiment: Robotic learning datasets and rt-x models: Open x-embodiment collaboration.
>
> [2] Toward hierarchical understanding of procedural activities.
>
> [3] Internvid: A large-scale video-text dataset for multimodal understanding and generation.
>
> [4] Videohallucer: Evaluating intrinsic and extrinsic hallucinations in large video-language models.
>
> [5] Evaluating object hallucination in large vision-language models.

---

> > ### Author Response · Authors · 2025-08-04
> >
> > Dear reviewer,
> >
> > We appreciate your time and effort in evaluating our work. As the discussion stage is ending soon, we wonder if our response answers your questions and addresses your concerns? Thanks again for your very constructive and insightful feedback!

---

> > > ### Comment · Reviewer_Fem7 · 2025-08-04
> > >
> > > I appreciate the authors’ response and explanations. My concerns, particularly regarding Data Quality Assurance, have been addressed. Based on the overall contribution, I maintain a rating of 4.

---

### Official Review · Reviewer_HXga · 2025-07-02

**Clarity:** 4
**Significance:** 4
**Originality:** 3
**Rating:** 5
**Confidence:** 4

**Summary:**

This paper introduces EgoThinker, a new framework designed to equip multimodal large language models (MLLMs) with strong egocentric video reasoning capabilities. Unlike traditional models that struggle with first-person understanding, EgoThinker combines spatio-temporal chain-of-thought (CoT) supervision and a two-stage learning curriculum. The authors first create EgoRe-5M, a large-scale dataset containing 13 million egocentric video clips with detailed QA annotations, CoT rationales, and hand–object grounding. EgoThinker is trained through supervised fine-tuning on EgoRe-5M and further refined with reinforcement learning to improve spatio-temporal localization. Experiments demonstrate that EgoThinker significantly outperforms prior approaches on egocentric benchmarks, especially in fine-grained reasoning and localization tasks.

**Questions:**

My major concern is that ST-Think tackles a very similar problem with a comparable training recipe; could you explicitly delineate how EgoRe-5M, your curriculum, or your evaluation protocol differs from ST-Think, and add the appropriate citations so that readers can clearly see the unique contributions of EgoThinker?

**Ethical Concerns:**

["NO or VERY MINOR ethics concerns only"]

**Final Justification:**

This paper makes a compelling contribution with EgoRe-5M, a large-scale and richly annotated dataset that fills a critical gap in egocentric video reasoning. The proposed two-stage training pipeline effectively combines supervised and reinforcement learning to align causal reasoning with spatial-temporal grounding. The model achieves state-of-the-art performance on multiple egocentric QA benchmarks while maintaining generalization across standard video tasks. Although there is some overlap with concurrent work, ST-Think, the authors highlighted the unique contributions of their dataset and compared the methodological differences between the two approaches in the rebuttal.

Given the significance of the dataset and the demonstrated effectiveness of the approach, I recommend this paper for acceptance.

**Limitations:**

Yes

**Quality:**

4

**Strengths And Weaknesses:**

Pros:
1. Rich, purpose-built dataset. EgoRe-5M offers 5 million QA pairs drawn from 13 million egocentric clips, with chain-of-thought rationales plus dense hand–object and temporal grounding, giving the community the first truly large-scale resource that couples reasoning and fine-grained localization.

2. Effective two-stage learning pipeline. The combination of supervised fine-tuning on EgoRe-5M and GRPO-based reinforcement fine-tuning tightly links high-level causal reasoning with low-level spatial–temporal grounding, as confirmed by ablation studies.

3. State-of-the-art egocentric performance. EgoThinker sets new highs on six egocentric QA benchmarks and more than doubles mIoU on EK-Visor relative to strong baselines, proving its practical impact on both reasoning and localization tasks.

4. Broad generalization. Despite its specialization, the model maintains or even slightly improves scores on standard video-understanding suites such as MVBench and Perception Test, indicating the method scales beyond egocentric data

Cons:

1. Novelty overlap. The core idea of pairing an egocentric QA dataset with chain-of-thought supervision and GRPO fine-tuning closely parallels the contemporaneous “ST-Think” work [1], which also introduces an ego-centric benchmark and an RL-enhanced MLLM.

[1]. Wu, P., Liu, Y., Liu, M. and Shen, J., 2025. St-think: How multimodal large language models reason about 4d worlds from ego-centric videos. arXiv preprint arXiv:2503.12542.

---

> ### Author Rebuttal · Authors · 2025-07-31
>
> Dear Reviewer,
>
> We sincerely thank you for your positive feedback to our work.
>
> Our work is differentiated from ST-Think by two fundamental aspects: our data and our methodology.
>
> 1.Data field. Our work introduces EgoRe-5M, a large-scale, richly annotated dataset created specifically to train models for egocentric reasoning. In contrast, ST-Think's principal contribution is a benchmark designed to evaluate the reasoning capabilities of existing models.
> Our efforts to build this training dataset include two novel contributions. First, we developed a novel data sourcing pipeline to automatically filter and curate egocentric video clips from large-scale web data. This directly addresses the critical scarcity of egocentric video sources for training. Second, we have several QA annotation types with a comprehensive annotation scheme to specifically enhance understanding, grounding, and reasoning abilities within egocentric contexts.
>
> 2.Method.
> While both studies use a two-stage training framework, our implementations and objectives at each stage are substantially different. SFT Stage: We strategically integrate our specialized EgoRe-5M data with general-purpose data. This approach is explicitly designed to enhance the model's expert egocentric reasoning skills while simultaneously preserving its broad, general-purpose capabilities. RFT Stage: Our reward functions target different objectives. ST-Think insightfully uses AI-generated scores to refine the model's general reasoning process. In contrast, our RFT stage is tailored to a core challenge in egocentric vision: understanding hand-object interactions. We formulate a reward signal that leverages limited grounding data to specifically sharpen the model's comprehension of these critical interactions, a focus essential for egocentric scenarios.
>
> In summary, while ST-Think provides a valuable benchmark for evaluation, our work introduces the data, resources, and a tailored methodology for training robust egocentric models. We agree that ST-Think offers several insightful ideas, and we will add a detailed discussion and citation in our revised manuscript to properly contextualize our contributions. Thank you again for your constructive feedback. We trust this clarifies the unique aspects of our work and addresses your concerns.
>
> We appreciate the reviewer pointing us to ST-Think, and we would like to respectfully note that this is a concurrent work with our submission, as it was released publicly as pre-prints very recently. Therefore, while it tackles a very similar problem in the field, our work was developed independently and made independent contributions in both data and method for egocentric reasoning.
> If you have any additional questions or suggestions, please feel free to tell us.

---

> > ### Author Response · Authors · 2025-08-04
> >
> > Dear reviewer,
> >
> > We appreciate your time and effort in evaluating our work. As the discussion stage is ending soon, we wonder if our response answers your questions and addresses your concerns? Thanks again for your very constructive and insightful feedback!

---

> > ### Comment · Reviewer_HXga · 2025-08-06
> > **thanks for your response**
> >
> > The rebuttal has addressed all of my concerns. I will keep my current score.

---

### Official Review · Reviewer_vznn · 2025-07-03

**Clarity:** 3
**Significance:** 2
**Originality:** 2
**Rating:** 3
**Confidence:** 4

**Summary:**

EgoThinker introduces a two‑part contribution to egocentric video reasoning.
1. **EgoRe‑5M dataset** – 5 million question‑answer pairs source from HowTo100M and existing EGO dataset spanning short‑term/long‑term/chain‑of‑thought/fine‑grained spatial/temporal grounding four main tasks.
2. **Two‑stage training pipeline** – SFT on EgoRe‑5M followed by GRPO fine‑tuning on the grounding split.
The resulting 7 B‑parameter model outperforms prior multimodal LLMs on egocentric benchmarks while matching general‑video performance.

**Questions:**

As the point to differentiate the paper is egocentric reasoning, the following questions are most related to the reasoning aspect of the paper.

1. Justification of reasoning
While it is acknowledged that certain egocentric tasks require varying degrees of reasoning, the paper does not discuss which types of queries benefit from what types of reasoning paths, nor does it justify whether such reasoning is necessary or accurate for each case. Please provide several fully worked examples spanning different query types, and clarify whether all long-horizon questions require a CoT rationale, or if the model can skip CoT when a direct answer suffices. Additionally, please comment on whether the DeepSeek V3-generated reasoning paths are effective for solving these queries.

2. Your current experiments stop at a 7B-parameter backbone, yet recent CoT literature (When More is Less: Understanding Chain-of-Thought Length in LLMs, Small Models Struggle to Learn from Strong Reasoners) shows that stronger reasoning benefits often surface only at larger scales. Could you report results for at least another series model—say, InternVideo 2.5-8 B to verify that the CoT-induced gains persist or even strengthen?

3. Hallucination and factual alignment audit
Because the CoTs are generated by a language-only DeepSeek LLM that never “sees” the video, there is a risk that rationales introduce objects or events absent from the footage. Could you (a) evaluate on an existing hallucination benchmark such as VQ2A-Hallucination or Image-CoT-Faithfulness. Presenting these numbers would let readers judge whether CoT length correlates with fidelity.

4. Grounding relevance
Egocentric frames are often cluttered with irrelevant objects. Please include an ablation study where the grounding reward is either removed or down-weighted:
    • Does rationale faithfulness (as measured above) improve or degrade?
    • Are certain question categories (e.g., action anticipation vs. spatial localization) more sensitive to grounding?
These results would help clarify whether grounding genuinely supports reasoning or simply provides an additional supervision signal.

Addressing these questions would substantially strengthen the paper, as many of the practices employed are standard in the field and offer limited novelty.

**Ethical Concerns:**

["NO or VERY MINOR ethics concerns only"]

**Limitations:**

Same in the Questions section.

**Quality:**

2

**Strengths And Weaknesses:**

**Strengths:**
- Extensive experiments demonstrate consistent state-of-the-art gains.
- Paper is well-structured with clear figures for pipeline, data statistics, and qualitative results.
- EgoRe-5M dataset will likely become a community resource.

**Weaknesses:**
- Chain-of-Thought justification: The paper does not clarify to what extent perception queries require CoT reasoning or what formal structure is assumed, only a few illustrative examples are provided.
- Hallucination / misalignment risk: CoTs are produced by a language-only DeepSeek model that never “sees” the video, so hallucinations are possible. No benchmark or audit quantifies factual alignment.
- CoT reliability: Longer rationales often amplify hallucinations, the paper adds more/longer CoTs but provides no hallucination benchmark (even an image-based one).
- Grounding relevance: Reward encourages pointing to any object, yet egocentric frames are cluttered. The authors do not show that grounding improves CoT quality or downstream accuracy, nor do they justify which objects matter.
- All annotations are synthetic, human validation is limited to a 500-sample spot-check.

---

> ### Author Rebuttal · Authors · 2025-07-31
>
> Due to the word limit, we will simplify the description in the rebuttal.
>
> ### **[Q1,W1] CoT data justification and egocentric reasoning cases.**
> We thank the reviewer for this insightful question.
> For cot data justification, We analyze from two perspectives:
>
> 1.Additional Benchmarks:
> We test EgoThinker on 2 additional benchmarks designed to measure complex reasoning, not just perception: ERQA[1] and VidEgoThink[2].
> | Model    | ERQA | VidEgoThink(Reward Model) |
> |---------|---------|---------|
> | Qwen2-VL | 37.0 | 48.8 |
> | EgoThinker | 41.8 | 51.7 |
> | Gemini1.5 | 41.8 | - |
>
> ERQA, proposed by Gemini Robotics, has various reasoning types such as spatial, trajectory and action reasoning. As shown in the table, our model's performance is comparable to Gemini 1.5, surpassing the baseline by 4.8%. For tasks requiring the model to infer whether an action was completed or understand long-term information (VidEgoThinker and QAEgo4D), EgoThinker's performance on both benchmarks shows our model's superior performance in reasoning tasks.
>
> 2.Examples of Egocentric Reasoning:
> The following examples, derived from our dataset, illustrate why a direct-answer approach is insufficient for these query types. As we are unable to upload images, so we only present QA examples.
>
> (1) Task Planning:
> Naturally, solving this problem requires breaking down tasks and providing a reasoning path based on logical relationships.
>
> 【Q】: How to add mortar mix and cement into the concrete mixer?
>
> 【A】: Pick up a sack of mortar mix with left hand while using right hand to hold tools. Then open the sack by tearing the seal with left hand while the right hand steadies the bag. Afterward, pour both the mortar mix and cement into the concrete mixer.
>
> 【Why CoT is necessary】: Without reasoning, the outputs is "You should pick up mortar mix and cement, and pour them into the concrete mixer." This is clearly insufficient since it lacks the actionable, sequential details.
>
> (2) Complex Event & State Inference:
> This task requires combining multiple implicit cues for accurate judgment; relying solely on direct observation without reasoning can lead to incorrect answers.
>
> 【Q】: Did the person complete the vegetable washing process?
>
> 【A】: The man put the vegetables into the basin and turned on the faucet, but then the camera moved elsewhere. However, when the camera returned to the basin, the man had turned off the faucet and removed his hands from the basin. Therefore, it can be inferred that he completed washing the vegetables.
>
> 【Why CoT is necessary】:In the scenario, we can't directly determine if the task is complete. However, by reasoning through the man's actions, camera movements, and the logical sequence, we can arrive at the answer.
>
> 3.Embodied reasoning: This requires combining spatial understanding with the context of an action.
>
> 【Q】:How do you need to rotate the dumbbell for it to fit back in the weight holder?
>
> 【A】:'To determine how to rotate the dumbbell for it to fit back in the weight holder, we first need to understand the orientation of the dumbbell and the weight holder.\n\n1. **Grounding of the Key Object**: The dumbbell is the key object here. It is shown lying on its side with the handle pointing upwards. The weight holder is positioned above the dumbbell, with its base facing downwards.. **Current Position**: The dumbbell is currently lying horizontally on the floor, with its handle pointing upwards. The weight holder is positioned above it, ready to receive the dumbbell.
> Thus you need to rotate dumbbell counter-clockwise 90 degrees and move it above to fit back in the weight holder.
>
> 【Why CoT is necessary】The answer depends on a multi-step mental process: (1) Identify the spatial information of the dumbbell and the target holder. (2) Calculate the necessary transformation. A direct perceptual guess is likely to fail.
>
> Furthermore, in scenarios involving camera transitions, spatial understanding, and motion understanding, integrating these pieces of information for reasoning is vital. However, most existing egocentric benchmarks focus on understanding and simple reasoning. Developing more complex egocentric reasoning benchmarks is a promising direction for future exploration.
>
> Regarding whether CoT is always required, the answer is no. As described in the paper, we first assess if a video segment was complex enough to warrant a CoT. For simpler queries, it is not generated or enforced.
>
> ### **[Q3,Q5,W2] Experiments on hallucination detection.**
> We thank the reviewer for raising this critical point regarding the risk of hallucination. Since the CoT rationales are generated from text captions without direct video access, ensuring factual alignment is paramount. We acknowledge this challenge.
>
> We've researched Hallucination detection and found that most benchmarks to test models for hallucinations rather than verifying if captions contain hallucinations. Methods like VQ2A-Hallucination or Image-CoT-Faithfulness aren't open-source. We also tried using the latest FClip, and ALOHa for verification, but these methods primarily evaluate image hallucination, and weren't suitable for our complex semantic and temporal information. Therefore, we conduct a 4-part audit of our EgoRe-5M CoT data to quantify its factual fidelity.
>
> 1.Annotation Process.
> As detailed in the paper, the CoT data was generated by DeepSeek-R1, a model with strong instruction-following capabilities. Our prompt explicitly constrained the model to generate rationales only from the information present in the source video caption, strictly forbidding the introduction of external or imagined details. This largely guarantees that the generated CoT won't contain hallucination.
>
> 2.CLIP-based Alignment Quality Analysis. We analyze the alignment quality between the caption and the video via the CLIP score. Specifically, we chose ViCLIP[3] pre-trained on a large amount of video data, and compared it against the original concatenated captions as a baseline. We found that the score of our CoTs is 62.3, while the original caption score is 70.6.
>
> While the CoT score is lower than the descriptive caption baseline, this is an expected outcome. The source captions describe concrete, visible objects and actions, which aligns perfectly with CLIP's training objective. In contrast, our CoT rationales articulate abstract reasoning steps, causal links, and inferred goals (e.g., “to align the dumbbell, a rotation is needed”), which do not always have a direct visual correlate. A score of 62.3 still indicates strong semantic alignment, confirming that the rationales are highly relevant to the video's content.
>
>
> 3.Manual Review.
> Based on the length of the generated CoT, we selected 5k samples from our CoT data split for manual review. The goal of this scoring was to assess whether the generated caption contained any non-existent objects or action descriptions when compared to the video. We set up a 4-point scale to categorize hallucination severity:
> Severe Hallucination, Moderate Hallucination, Partial Hallucination and No Hallucination.
> We get 4068, 577, 279, 76 for score = 3,2,1,0. We observe that hallucinations primarily manifest as the model outputting non-existent goals or actions based on the provided captions, with hardly any instances of non-existent objects. Furthermore, the majority of captions exhibit no hallucinations or only minor ones, indicating good alignment quality in our CoT generation.
>
> 4.To validate whether EgoThinker exhibits hallucinations, we conducted evaluations on two widely-adopted benchmarks: (1) VideoHallucer[4] for assessing large video-language models, which contains object-relation, temporal, semantic detail and extrinsic hallucination detection. (2) POPE (Polling-based Object Probing)[5] for object hallucination detection.
>
> | Model    | VideoHallucer | POPE |
> |---------|---------|---------|
> | Qwen2-VL | 47.6 | 83.6 |
> | EgoThinker | 47.0 | 86.8 |
> | GPT4o | 53.3| - |
>
> The experimental results demonstrate a marginal performance decrease (0.6%) on VideoHallucer, while achieving a significant improvement of 3.2% on the POPE benchmark. We attribute this differential performance to our model's enhanced object grounding capability, which particularly enhances its robustness against object hallucination.
>
> ### **[Q2] Experiments on another series model**
> We finetune the more powerful video MLLM InternVideo2.5-8B. The table below shows our fine-tuned results across various datasets. From the table, we can see that our dataset shows minor improvements on EgoTaskQA and QAEgo4D. We believe this is because InternVideo2.5-8B inherently possesses strong video understanding capabilities. However, on EgoSchema and ERQA, which are more dependent on reasoning and are more complex, InternVideo2.5 achieved more significant improvements.
>
> | Model    | EgotaskQA | QAEgo4D | EgoSchema | ERQA |
> |---------|---------|---------|---------|---------|
> | QwenVL2-VL-7B | 57.9 | 60.3 | 63.3 |37.0 |
> | QwenVL2-VL-7B+ours | 64.4 | 66.2 | 67.6 |41.8|
> | Internvideo2.5 | 70.3 | 67.0 | 63.9 | 35.6 |
> | Internvideo2.5+ours | 72.4 | 68.3 | 66.4 | 40.0 |
>
> Due to resource and time limitations, we haven't yet completed experiments with larger models, such as the 13B-level model. We will supplement these experiments in the future and further analyze the role of reasoning data.
>
> ### **[Q4] Grounding Data with Egocentric Video Understanding**
> Due to word limit, please refer to the **Relationship Between Grounding Data and Egocentric Video Understanding** section in our rebuttal for Reviewer k3DC(last).
>
> [1] Gemini robotics: Bringing ai into the physical world.
>
> [2] Videgothink: Assessing egocentric video understanding capabilities for embodied ai.
>
> [3] Internvid: A large-scale video-text dataset for multimodal understanding and generation.
>
> [4] Videohallucer: Evaluating intrinsic and extrinsic hallucinations in large video-language models.
>
> [5] Evaluating object hallucination in large vision-language models.

---

> > ### Author Response · Authors · 2025-08-04
> >
> > Dear reviewer,
> >
> > We appreciate your time and effort in evaluating our work. As the discussion stage is ending soon, we wonder if our response answers your questions and addresses your concerns? Thanks again for your very constructive and insightful feedback!

---

### Note · Authors · 2025-08-12

We sincerely thank the reviewers and the AC for their insightful feedback and constructive suggestions, and we have carefully addressed all the concerns raised during the rebuttal phase.

First, we would like to summarize the strengths of our work as highlighted by the reviewers:

1. EgoRe-5M: A large-scale dataset for egocentric reasoning that spans multiple tasks and is poised to become a valuable community resource.

2. Effective pipeline: A two-stage SFT+RFT framework delivering significant improvements in fine-grained grounding while preserving general video understanding.

3. Extensive experiments: Comprehensive evaluations demonstrating state-of-the-art results on several tasks including reasoning and grounding tasks.

Then, we summarise the core concerns raised by the reviewers and our solutions, which will significantly strengthen our work:

1. **Data Quality Justification**: We have further validated the quality of our data from **four perspectives**: clip-based score, hallucination detection, manual review, and a clarification of our data processing pipeline. **(For vznn, k3DC, Fem7)**

2. Analysis of **the relation between grounding/reasoning data and egocentric video understanding**: We analyze the deficiencies of existing egocentric benchmarks in evaluating fine-grained and complex reasoning capabilities. To demonstrate the significance of our data, we select more challenging benchmarks(ERQA, VidEgothink) for validation. Furthermore, we have provided multiple instances to illustrate the importance of grounding abilities for egocentric reasoning. **(For vznn, k3DC)**

3. Contributions of our training paradigm: We have further elaborated on our contributions in the stage of RFT. Specifically, we adapt **a single-task batching strategy**, which effectively addresses the slow convergence and suboptimal performance when simultaneously utilizes RFT for multiple tasks. **(For HXga, k3DC, Fem7)**

We appreciate that most reviewers have provided positive final feedback after considering our rebuttal. While Reviewer vznn did not participate in the discussion stage, we have comprehensively addressed the concerns on data quality, CoT utility, and grounding–reasoning connections with additional analyses and experiments in our rebuttal. We trust that in the final discussion stage a fair evaluation will be made based on the latest feedback and our clarifications.

---

### Decision · Program_Chairs · 2025-09-17

**Decision:**

Accept (poster)

**Comment:**

The submission initially received mixed ratings, with two reviewers leaning toward rejection (vznn, k3DC) and two toward acceptance (HXga, Fem7). Reviewers appreciated the introduction of a useful dataset (EgoRe-5M), the two-stage training pipeline with comprehensive ablations, and the demonstrated state-of-the-art performance. Concerns were raised, however, about the impact of hallucinations in the CoT process and the degree of novelty given overlaps with existing work (e.g., ST-Think, Robobrain, UniVG-R1).

The authors provided an effective rebuttal. Reviewer k3DC subsequently raised their score to borderline accept, while the reviewers already leaning toward acceptance maintained their positions. Although vznn did not update their score or provide post-rebuttal feedback, the AC sees that the rebuttal addressed their concerns regarding hallucinations and the justification of CoT.

Overall, the AC agrees with the majority of reviewers in recommending acceptance. While the contribution is moderate, the paper introduces a valuable dataset and an effective two-stage approach that achieves strong performance. The authors are encouraged to incorporate the rebuttal into the final version, particularly the discussion on hallucinations and comparisons with related work.